

# Rotation of floating particles in submesoscale cyclonic and anticyclonic eddies: a model study for the southeastern Baltic Sea

Victor Zhurbas[1], Germo Väli[2], Natalia Kuzmina[1]

[1]Shirshov Institute of Oceanology, Russian Academy of Sciences, 36 Nakhimovsky Prospect, 117997 Moscow, Russia

[2]Tallinn University of Technology, Department of Marine Systems, Akadeemia tee 15A, 12618 Tallinn, Estonia

*Correspondence to*: Victor Zhurbas (zhurbas@ocean.ru)

## Abstract

It was assumed that the overwhelming dominance of cyclonic spirals on satellite images of the sea surface could be caused by some sort of differences between rotary characteristics of the submesoscale cyclonic and anticyclonic eddies. This hypothesis was tested by means of numerical experiments with synthetic floating Lagrangian particles embedded in a regional circulation model of the southeastern Baltic Sea with very high horizontal resolution (0.125 nautical mile grid). The numerical experiments showed that the cyclonic spirals can be formed both from a horizontally uniform initial distribution of floating particles and from the initially lined up particles during the advection time of the order of 1 day. Statistical processing of the trajectories of the synthetic floating particles allowed to conclude that the submesoscale cyclonic eddies differ from the anticyclonic eddies in three ways favouring the formation of the spirals: the former can be characterized by a considerably higher angular velocity and a more pronounced differential rotation as well as by a negative helicity.

Keywords: submesoscale eddies; cyclonic spirals; Baltic Sea; numerical modelling; satellite imagery.



# 1 Introduction

Spiral structures that can be treated as signatures of submesoscale eddies are a common feature on the synthetic aperture radar (SAR), infrared, and optical satellite images of the sea surface (e.g. Munk et al., 2000; Laanemets et al., 2011; Karimova et al., 2012; Ginzburg et al., 2017). The spirals are broadly distributed in the World Ocean, 10—25 km in size and overwhelmingly cyclonic (Munk et al., 2000). Walter Munk (Munk, 2001) has summarized a formation mechanism of the spirals as follows: "*Under light winds favorable to visualization, linear surface features with high surfactant density and low surface roughness are of common occurrence. We have proposed that frontal formations concentrate the ambient shear and prevailing surfactants. Horizontal shear instabilities ensue when the shear becomes comparable to the Coriolis frequency. The resulting vortices wind the linear features into spirals.*". Horizontal shear instabilities were shown to favour cyclonic shear and cyclonic spirals for different reasons (Munk et al., 2000). Note that the submesoscale flows are the upper ocean layer flows with horizontal length scale of the order of 0.1−10 km that are characterized by the Rossby number (the ratio of relative vertical vorticity to the Coriolis frequency) and the Richardson number (the ratio of the squared buoyancy frequency to the squared vertical shear) of the order of unity, as well as by a conspicuous asymmetry of the relative vertical vorticity distribution with a tail of enhanced positive (cyclonic) vorticity values (Thomas et al., 2008; McWilliams, 2016). Submesoscale processes play an important role in turbulence and mixing of the upper ocean layer (Fox-Kemper et al., 2008, 2011; Thomas et al., 2008; McWilliams, 2016). While horizontal shear or barotropic instability is one possible mechanism for generating submesoscale eddies (Munk's hypothesis), more recent studies have shown that the mixed-layer baroclinic instabilities (Haine and Marshall, 1998) are a more plausible explanation for the observed submesoscale vortices (e.g., Eldevik and Dysthe, 2002; Boccaletti et al., 2007; Dewar et al., 2015; Molemaker et al., 2015; Buckingham et al., 2017). Submesoscale structures and the associated instabilities were simulated using high-resolution circulation models in various areas of the World Ocean such as the California Current system (Capet et.al., 2008; Dewar et al., 2015; Molemaker at al., 2015), the Gulf Stream (Gula et al., 2016), the Gulf of Mexico (Barkan et al., 2017). Similarly,


high-resolution circulation models with the horizontal grid of less than 0.6 km were implemented also
to study submesoscale dynamics in the Baltic Sea (Vankevich et al., 2016; Väli et al., 2017, 2018;
Vortmeyer-Kley et al., 2019; Zhurbas et al., 2019; Onken et al., 2019).

To our mind the common occurrence of spirals on satellite images of the sea surface hints that the
winding of the linear features in the course of development of the horizontal shear instabilities and/or
the mixed-layer baroclinic instabilities is not the only way to generate the spirals. Rather one may
expect that the spirals can also be generated by the advection of a floating tracer in a velocity field
inherent to mature, relatively long-living submesoscale/mesoscale eddies, and the initial tracer
distribution is not necessarily characterized by the linear surface features. If it holds, then for the
predominance of cyclonic spirals over the anticyclonic spirals, some properties of the rotary motion of
floating particles, such as angular velocity, differential rotation and helicity, should be different for
cyclonic and anticyclonic eddies. The objective of this work is to assess the differences between
floating particles rotation in the submesoscale cyclonic and anticyclonic eddies, which can be
responsible for overwhelmingly cyclonic spirals in the satellite images, by means of a very high
resolution modelling as applied to the southeastern Baltic Sea.

Spirals in the southeastern Baltic Sea were repeatedly observed in infrared (e.g. Zhurbas et al., 2004;
Ginzburg et al., 2017), SAR (Karimova et al., 2012), and optical (e.g. Karimova et al., 2012; Ginzburg
et al., 2017) images. Most fabulous optical images have been encountered in summer when the spirals
become visualized by the cyanobacteria blooms. An example of a prominent cyclonic spiral located at a
distance of 60 km north-northwest from the Cape Taran visible on Landsat-8 optical image due to
cyanobacteria blooms is presented in Fig. 1. Note that the cyclonic spiral actually is a constituent of a
vortex pair consisting of coupled cyclonic and anticyclonic eddies, the latter located at about 30 km to
the south of the former. However, the anticyclonic eddy does not form a prominent spiral like the
cyclonic eddy. As it was mentioned above, a better visualization of the cyclonic spirals is supposedly
related to some differences between floating particles rotation in submesoscale cyclonic and
anticyclonic eddies which will be investigated hereafter.



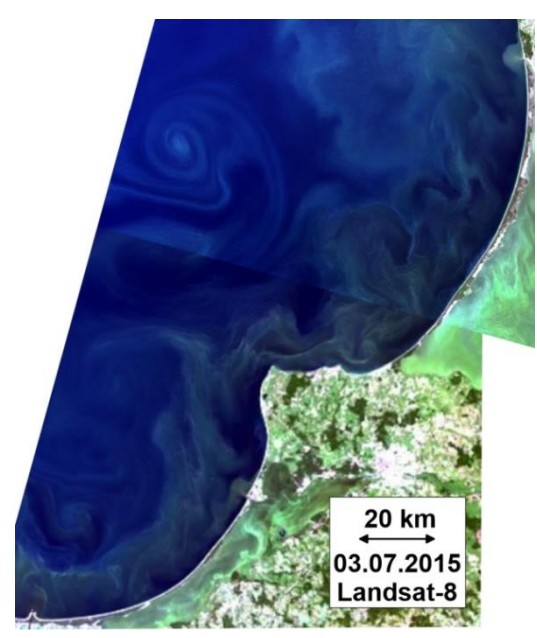


Fig. 1. Landsat-8 true color image of the southeastern Baltic Sea with a prominent cyclonic spiral located at a distance of about 60 km to the north-northwest from the Cape Taran. The image was downloaded from https://eos.com/landviewer on 24 June 2018, © Copyright 2019, EOS DATA ANALYTICS, Inc © OpenStreetMap contributors 2019. Distributed under a Creative Commons BY-SA
License.

## 2   Material and methods

### 2.1   Model setup

The General Estuarine Transport Model (GETM) (Burchard and Bolding, 2002) was applied to simulate the meso- and submesoscale variability of temperature, salinity, currents, and overall dynamics
in the southeastern Baltic Sea. GETM is a primitive equation, 3-dimensional, free surface, hydrostatic model with the embedded vertically adaptive coordinate scheme (Hofmeister et al., 2010; Gräwe et al., 2015). The vertical mixing is parametrized by two equation k-ε turbulence model coupled with an algebraic second-moment closure (Canuto et al., 2001; Burchard and Bolding, 2001). The implementation of the turbulence model is performed via General Ocean Turbulence Model (GOTM)
(Umlauf and Burchard, 2005).





The horizontal grid of the high-resolution nested model with uniform step of 0.125 nautical miles (approximately 232 m) all over the computational domain, which covers the central Baltic Sea along with the Gulf of Finland and Gulf of Riga (Fig. 2), was applied while in the vertical direction 60 adaptive layers were used. The digital topography of the Baltic Sea with the resolution of 0.5 nautical miles was obtained from the Baltic Sea Bathymetry Database (http://data.bshc.pro/) and interpolated to the resolution required.

The model simulation run was performed from 1 April to 9 October 2015. The model domain has the western open boundary in the Arkona Basin and the northern open boundary at the entrance to the Bothnian Sea. For the open boundary conditions the one-way nesting approach is used and the results from the coarse resolution model are utilized at the boundaries. The coarse resolution model covers the entire Baltic Sea with an open boundary in the Kattegat and has the horizontal resolution of 0.5 n.m. (926 m) over the whole model domain. More detailed information on the coarse resolution model is available in Zhurbas et al. (2018).

The atmospheric forcing (the wind stress and surface heat flux components) is calculated from the wind, solar radiation, air temperature, total cloudiness and relative humidity data generated by HIRLAM (High Resolution Limited Area Model) maintained operationally by the Estonian Weather Service with the spatial resolution of 11 km and temporal resolution of 1 hour (Männik and Merilain, 2007). The wind velocity components at the 10 m level along with other HIRLAM meteorological parameters are interpolated to the model grid.

The freshwater input from 54 largest Baltic Sea rivers together with their inter-annual variability is taken into account in the coarse resolution model. The original dataset consists of daily climatological values of discharge for each river, but inter-annual variability is added by adjusting the freshwater input to different basins of the sea to match the values reported annually by HELCOM (Johansson, 2018). The high-resolution model accounts only for rivers that flow into the sea within the model domain.

The initial thermohaline field was obtained from the coarse resolution model for 1 April 2015 and interpolated to the high-resolution model grid. The prognostic model runs were started from motionless state and zero sea surface elevation. The spin-up time of the southern Baltic Sea model under the





atmospheric forcing was expected to be within 10 days (Krauss and Brügge, 1991; Lips et al., 2016),
while the model output for comparison with the respective satellite imagery was obtained after 45 days

of simulation.

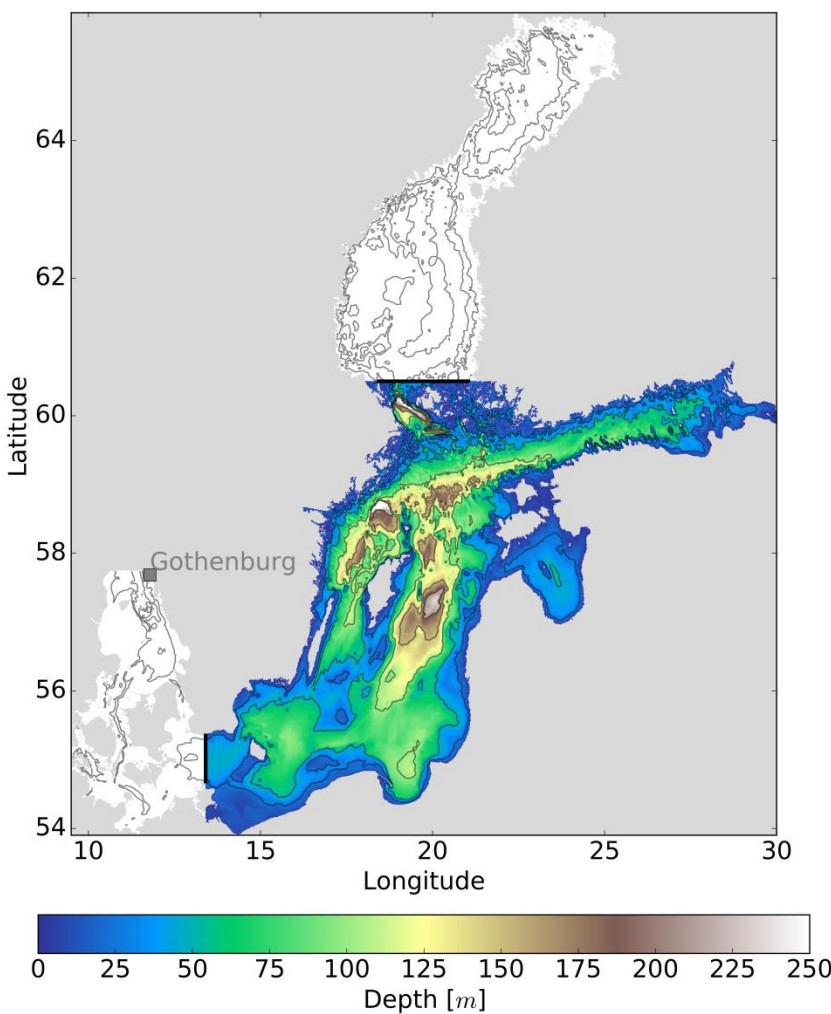

Fig. 2. Map of the high resolution model domain (filled colours) with the open boundary locations
(black lines). Coarse resolution model domain (blank contours + filled colours) has an open boundary

close to the Gothenburg station.


## 2.2 Application of synthetic floating particles approach to extract rotary characteristics of submesoscale cyclones/anticyclones

In order to characterize the submesoscale eddies, we estimated eddy radius $R$, the dependence of angular velocity of rotation $\omega(r)$ on radial distance from the eddy centre $r$, angular velocity in the eddy

centre $\omega_0 \equiv \omega(0)$, differential rotation parameter $Dif = [\omega(0) - \omega(R)]/\omega(0)$ and helicity parameter $Hel$, which will be defined later. The approach to calculate $\omega(r)$ and other parameters is illustrated in Fig. 3, where a pseudo-trajectory of a synthetic floating particle deployed within a modelled submesoscale eddy is presented. The pseudo-trajectory was calculated using a frozen velocity field, i.e. we took the modelled surface velocities for a given instant and kept the velocity field stationary during

the whole advection period.

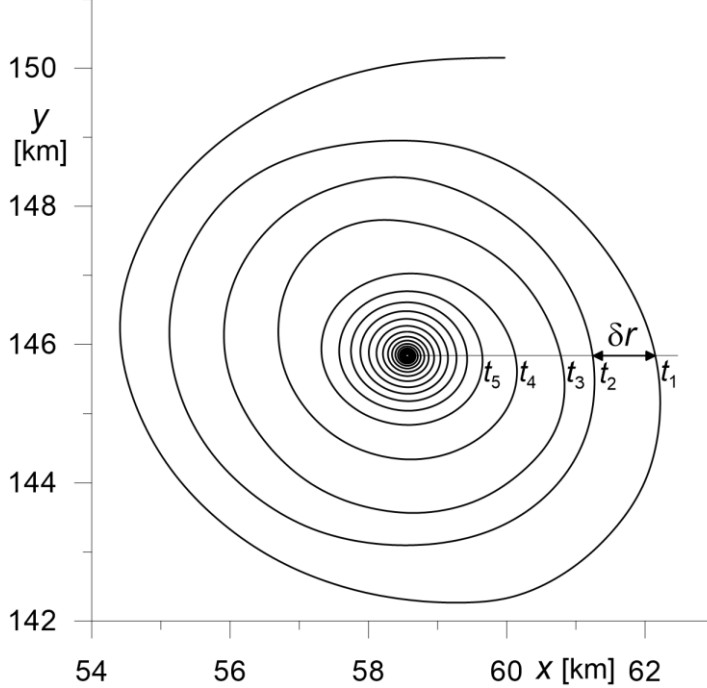

Fig. 3. An example of pseudo-trajectory of a synthetic floating particle deployed in a submesoscale eddy. The pseudo-trajectory was calculated using a surface velocity field in the southeastern Baltic Sea simulated for the time moment 15.05.2015, 12:00 (the frozen field approach). The particle was released

in the periphery of the submesoscale cyclonic eddy c1 (see Fig. 4).





If $t_1$ and $t_2$ are the start and end time of a full particle loop (see Fig. 3), respectively, then current values of $\omega$ and $r$ can be calculated as

$$\omega = 2\pi/(t_2 - t_1), r = l/(2\pi), \qquad (1)$$

where $l$ is the length of the pseudo-trajectory loop corresponding to the time interval $[t_1, t_2]$. Note that a plain linear relation between the vorticity $\zeta$ and the frequency of rotation in the axisymmetric eddy, $\zeta = 2\omega$, is valid only for the rigid-body rotation when $\omega(r) = const$, while for the differential rotation a more complicated formula is applied

$$\zeta = \frac{1}{r}\left[\frac{\partial}{\partial r}\left(rV_\varphi\right)\right] = \frac{1}{r}\left[\frac{\partial}{\partial r}(r^2\omega)\right] = 2\omega + r\frac{\partial\omega}{\partial r}, \qquad (2)$$

where $V_\varphi$ is the transversal component of velocity.

The helicity parameter can be introduced as

$$Hel = \frac{\delta r}{r}, \qquad (3)$$

where $\delta r$ is the change of $r$, either positive or negative, for the time interval $[t_1, t_2]$ (see Fig. 3). If $Hel \ll 1$ in an axisymmetric eddy, it can be presented as $Hel = 2\pi V_r/V_\varphi$, where $V_r$ is the radial component of velocity. Deploying synthetic floating particles at different distance from the eddy centre and applying approach (1)–(3), one can build functions $\omega(r)$ and $Hel(r)$. If a particle is deployed at a large enough distance from the eddy centre, the pseudo-trajectory will inevitably cease to be looped, and the largest $r$ calculated from a still loop-shaped trajectory is taken for eddy radius $R$. Once $\omega(r)$, $Hel(r)$ and $R$ are calculated, one can assess differential rotation $Dif$, mean helicity parameter $\langle Hel \rangle$ as well as angular velocity in the eddy centre $\omega_0$ as

$$Dif = \frac{[\omega(0) - \omega(R)]}{\omega(0)}, \langle Hel \rangle = \frac{1}{R}\int_0^R Hel(r)dr, \omega_0 = \omega(0). \qquad (4)$$

Instead of $\omega_0$ we used normalized frequency of rotation in the eddy centre $\Omega_0 = 2\omega_0/f$, where $f$ is the Coriolis frequency. Note that $Hel(r)$ is, in principle, an alternating function which proves the necessity of its averaging to get the bulk value $\langle Hel \rangle$. The positive (negative) value of $\langle Hel \rangle$ manifests the divergence (convergence) of currents and the related upwelling (downwelling) in the surface layer of the eddy.





It can be easily seen that the large value of $Dif$ and $\omega_0$ and the negative value of $Hel(r)$ favour the formation of spirals from linear features. Indeed, if $Dif = 0$ (solid body rotation) the linear feature within the eddy will remain linear but rotated by some angle relative to the initial position (i.e. no spiral pattern is formed), whereas a positive $\langle Hel \rangle$ will result in sweeping the particles out from the eddy core, thus making the spiral less visible. And the large value of $\omega_0$ will accelerate the formation of the spiral, provided that $Dif$ is large enough and $\langle Hel \rangle$ is negative (or sufficiently small positive). Since the spirals are known to be overwhelmingly cyclonic, one may expect that $Dif$ and $\omega_0$ will be larger and $\langle Hel \rangle$ will be smaller for the submesoscale cyclonic eddies relative to those for the anticyclonic eddies.

Apart from the above defined rotary characteristics of submesoscale eddies calculated from frozen velocity field, we addressed some numerical experiments with the deployment of synthetic floating particles in the modelled non-stationary (not frozen) velocity field, namely, when initially the particles were uniformly distributed on the sea surface, and when initially the particles formed a linear feature (i.e. a line) passing through the centre of a cyclonic or anticyclonic eddy.

The trajectories of floating particles were calculated by means of numerical integration of plain equations of the Lagrangian particle advection with a Runge-Kutta scheme of higher order of accuracy (Väli et al., 2018).

# 3   Results

## 3.1   Modelled submesoscale fields of surface velocity and temperature in comparison with satellite imagery

Modelled snapshots of surface layer temperature and currents with submesoscale resolution in the southeastern Baltic Sea for 15 May, 8 June and 3 July 2015, are shown in Figs. 4–6, respectively. The snapshots demonstrate a quite dense packing of the sea surface with submesoscale eddies. Similar dense packing of the sea surface with submesoscale eddies was observed in Envisat ASAR WSM images of the southeastern Baltic Sea (Karimova et al., 2012). Looking at the snapshots of the surface layer currents (panels (b) in Figs. 4–6), one cannot see any predominance of the number of cyclones over the number of anticyclones or vice versa. However, the surface layer temperature snapshots (panels (a) in




Figs. 4–6), clearly demonstrate a large number of spiral structures linked with the submesoscale cyclonic eddies, while the submesoscale anticyclones, as a rule, do not manifest themselves by well-defined spirals.

Some of the simulated submesoscale eddies shown in Figs. 4–6 were chosen for further calculations of their rotary characteristics by means of the approach described in Chapter 2.2. In total, the calculations were performed for 18 anticyclonic and 18 cyclonic eddies marked in Figs. 4–6, panels (b) as a1–a18 and c1–c18, respectively. The results are presented in Chapter 3.4.

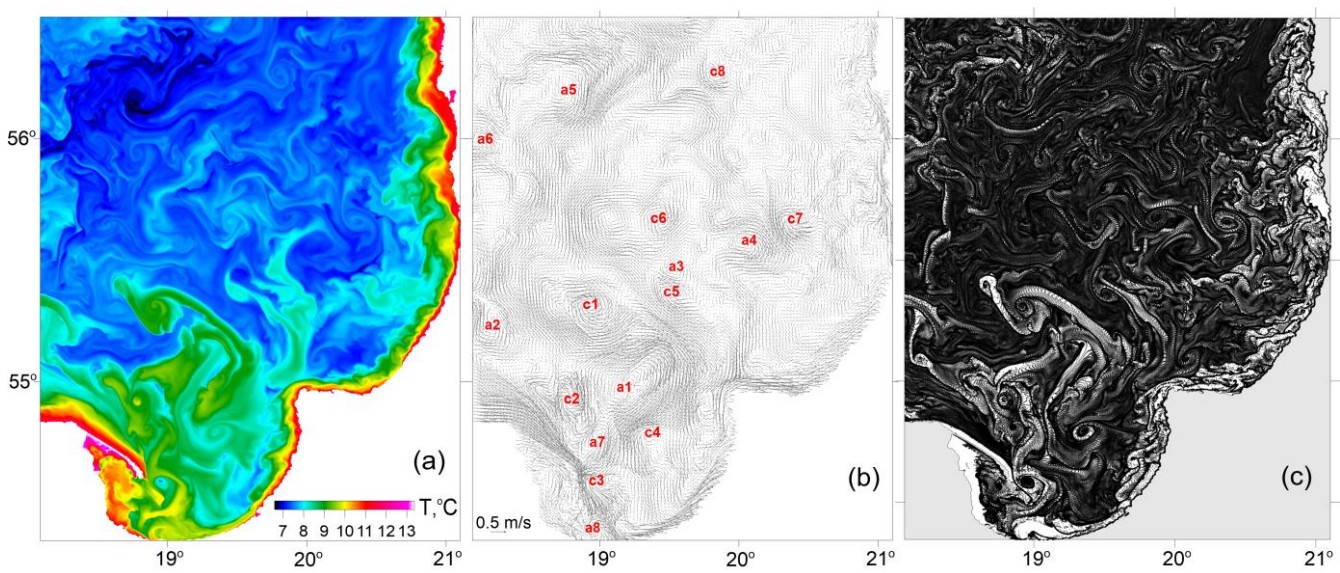

Fig. 4. Modelled fields of the surface layer parameters in the southeastern Baltic Sea on 15 May 2015: temperature (a), current velocity (b), and spatial distribution of uniformly released synthetic floating Lagrangian particles (c) after 1 day of advection. The red labels in panel (b) point at cyclonic (c1, c2, etc.) and anticyclonic (a1, a2, etc.) eddies used to calculate rotary characteristics in Chapter 3.4 (see Table 1).

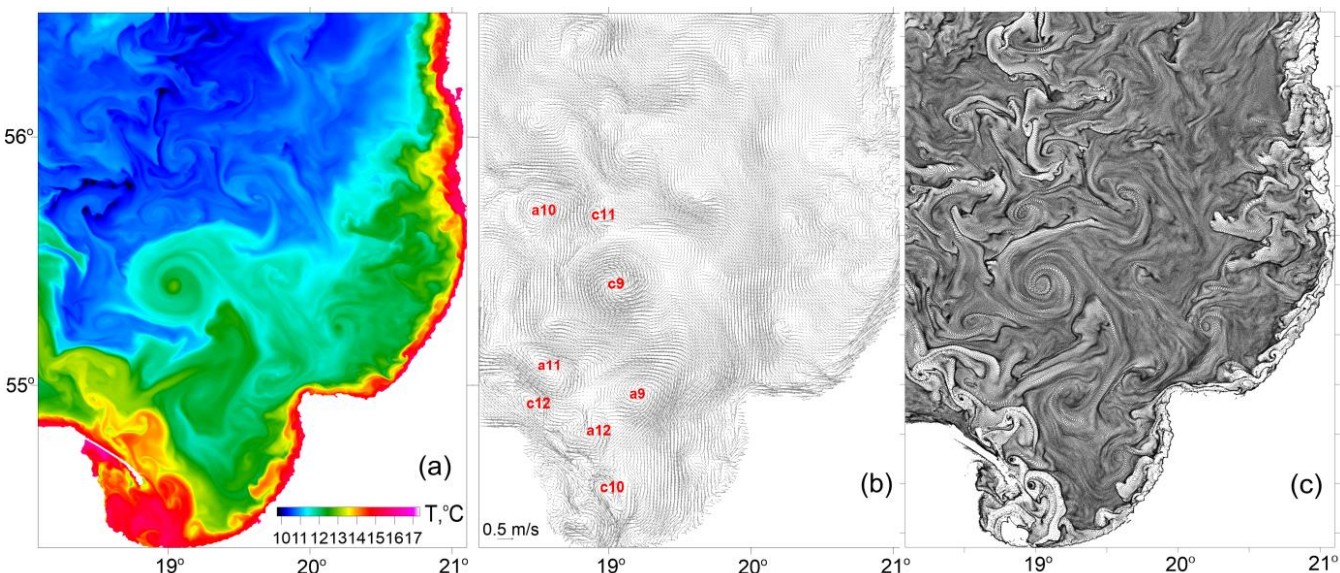


Fig. 5. The same as in Fig. 4 but for the date of 08.06.2015.

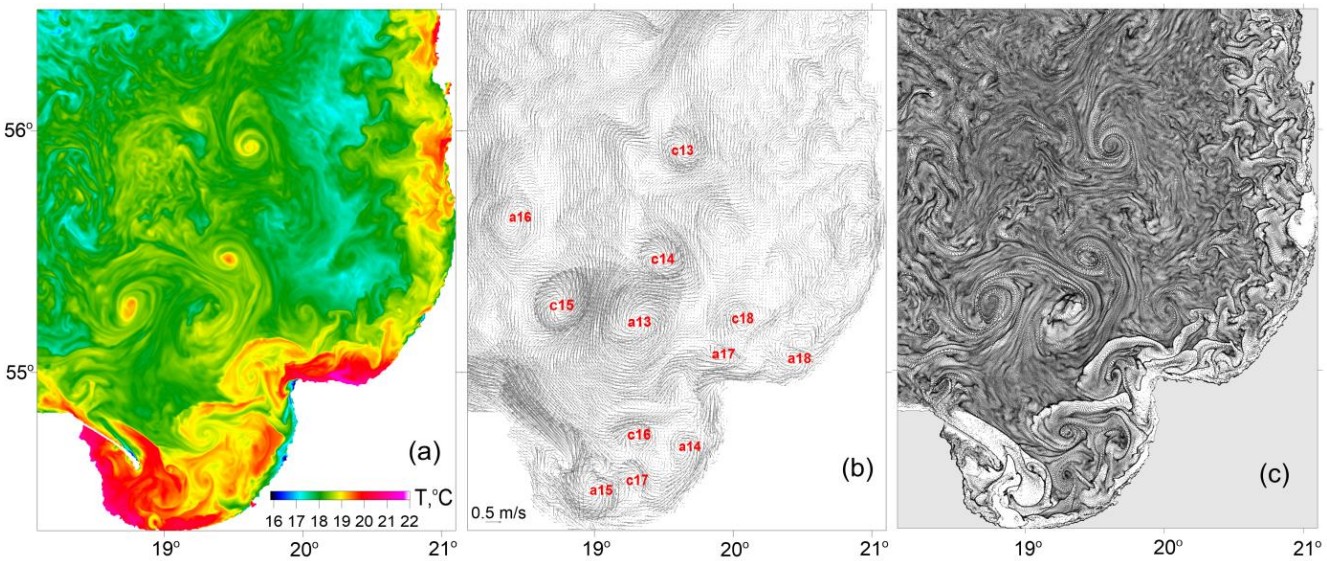

Fig. 6. The same as in Figs. 4 and 5 but for the date of 03.07.2015.

Note that the modelled snapshots of surface layer temperature and currents presented in Fig. 6

correspond to the date 03.07.2015, for which we have a true colour image of the southeastern Baltic Sea

from Landsat-8 (Fig. 1). A vortex pair seen in the satellite image at the distance of 30−60 km northwest



from the Cape Taran can be also identified in the simulated temperature and current fields of the surface layer; it is labelled as c14 and a13 in Fig. 6b. Moreover, to the south from the vortex pair c14–a13 in the Gulf of Gdansk, both the model and the satellite image display 2–3 cyclonic eddies (cf. Figs. 1 and 6).

The possibility to identify the observed vortex pair in the simulated fields can be considered as a validation of the model.

## 3.2 Numerical experiments with spatially uniform release of synthetic floating particles

Patterns formed on the sea surface by synthetic floating Lagrangian particles were shown to be a

powerful tool to visualize the mesoscale/submesoscale structures (Väli et al., 2018). Examples of such patterns are also presented in Figs. 4–6, panels (c). The particles were deployed uniformly (i.e. one particle in the centre of the every grid bin, the total number of particles was approx. 1 million) within the model domain a day before the date specified in Figs. 4–6 and carried by the simulated nonstationary currents during 1 day (i.e. $\tau = 1$ day, where $\tau$ is the advection time). Soon after the

release of synthetic floating particles, the horizontally uniform distribution of particles was transformed into a pattern that resembles the corresponding maps of oceanographic tracers such as temperature and/or salinity in the surface layer. Therefore, the floating particles allow easily visualize submesoscale structures. Note, that within just one day of advection the uniformly distributed particles clustered predominantly into cyclonic spirals corresponding to submesoscale eddies.

## 235   3.3 Numerical experiments with linearly aligned release of synthetic floating particles in submesoscale cyclones/anticyclones

Keeping in mind that according to Munk et al. (2000) the spirals can be formed from linear surface features winded by vortices, numerical experiments were performed with synthetic floating particles initially clustered in zonally aligned features intersecting the centres of the submesoscale cyclones

marked as c13–c18, and anticyclones marked as a13–a16 and a18 in Fig. 6. Figure 7 shows the





evolution of a linear feature of a large number of synthetic floating particles in 1 and 2 days of advection in the simulated velocity field.

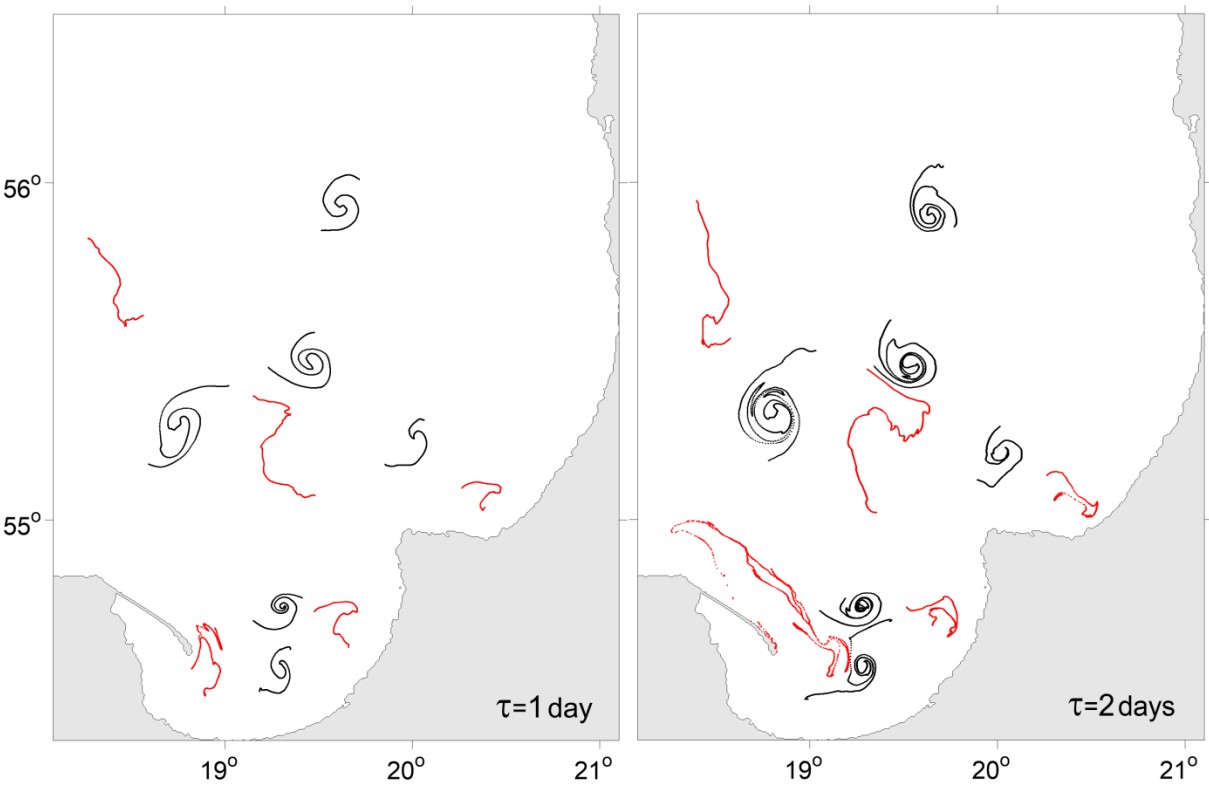

Fig. 7. Patterns formed in 03 July 2015 from zonally elongated linear features passing through the centres of the simulated submesoscale cyclonic (black curves) and anticyclonic (red curves) eddies after one (left) and two (right) days of advection. The linear features included a large number (2000) of synthetic floating particles deployed a day (left) and two days (right) before 03 July 2015.

It is clearly seen from Fig. 7 that the spirals were formed only from the linear features embedded into the submesoscale cyclonic eddies, while the linear features in the anticyclonic eddies transformed to some curves of irregular shape.



## 3.4 Numerical experiments with the release of synthetic floating particles in a frozen velocity field to extract rotary characteristics of submesoscale cyclones/anticyclones

Applying the approach described in Chapter 2.2 rotary characteristics $R$, $\Omega_0 = 2\omega_0/f$, $Dif$ and $\langle Hel \rangle$ were calculated for 18 anticyclonic eddies and 18 cyclonic eddies (marked as a1–a18 and c1–c18, respectively, in Figs. 3–6, panels (b)). The rotary characteristics of individual eddies along with the mean values, standard deviations and 95% confidence intervals calculated for the anticyclones and cyclones separately are presented in Table 1. For clarity, the scatter plots of $R$, $Dif$ and $\langle Hel \rangle$ versus $\Omega_0$ are shown in Fig. 8.

Table 1. Rotary characteristics of submesoscale cyclonic and anticyclonic eddies.

| Eddy ID | $R$, km | $\Omega_0 = 2\omega_0/f$ | $\langle Hel \rangle$ | $Dif$ |
|---|---|---|---|---|
| a1 | 16.22 | -0.24 | 0.72 | 1.86 |
| a2 | 5.26 | -0.48 | 0.36 | 2.11 |
| a3 | 7.72 | -0.40 | 0.35 | 3.45 |
| a4 | 6.63 | -0.45 | 0.07 | 1.86 |
| a5 | 6.42 | -0.34 | 1.14 | 3.02 |
| a6 | 5.71 | -0.49 | 1.08 | 2.21 |
| a7 | 4.82 | -0.46 | 1.00 | 1.67 |
| a8 | 1.36 | -1.56 | -0.04 | 1.59 |
| a9 | 11.03 | -0.56 | -0.03 | 4.18 |
| a10 | 7.18 | -0.47 | -0.07 | 1.99 |
| a11 | 11.62 | -0.53 | 1.48 | 3.46 |
| a12 | 4.33 | -0.54 | 0.35 | 1.71 |
| a13 | 11.32 | -0.41 | 0.86 | 2.30 |
| a14 | 6.71 | -0.84 | 1.00 | 3.20 |
| a15 | 5.35 | -0.96 | 0.66 | 2.70 |
| a16 | 10.14 | -0.40 | 0.72 | 3.41 |
| a17 | 3.41 | -0.36 | -0.04 | -0.71 |
| a18 | 4.68 | -0.77 | 0.61 | 2.77 |
| a1–a18: mean | 7.22 | -0.57 | 0.57 | 2.38 |
| standard deviation | 3.60 | 0.31 | 0.48 | 1.08 |
| 95% conf. interval | [5.43, 9.01] | [-0.72, -0.42] | [0.33, 0.81] | [1.84, 2.92] |
| c1 | 4.67 | 1.67 | -0.42 | 2.95 |




| | | | | |
|---|---|---|---|---|
| c2 | 6.07 | 3.66 | 0.00 | 8.19 |
| c3 | 2.69 | 2.59 | 0.25 | 2.79 |
| c4 | 4.02 | 1.01 | 0.09 | 4.33 |
| c5 | 7.92 | 1.09 | 0.08 | 5.68 |
| c6 | 8.51 | 0.96 | -0.15 | 6.72 |
| c7 | 4.34 | 1.62 | 0.20 | 3.36 |
| c8 | 6.67 | 1.41 | -0.13 | 13.25 |
| c9 | 14.59 | 1.60 | 0.07 | 11.31 |
| c10 | 5.28 | 2.48 | 0.31 | 7.08 |
| c11 | 2.97 | 1.33 | -0.21 | 3.61 |
| c12 | 11.72 | 1.58 | -0.10 | 10.20 |
| c13 | 7.90 | 1.30 | -0.06 | 9.84 |
| c14 | 6.86 | 1.43 | 0.20 | 3.60 |
| c15 | 9.04 | 1.60 | 0.18 | 5.16 |
| c16 | 4.96 | 1.85 | -0.56 | 4.58 |
| c17 | 3.82 | 1.30 | -0.38 | 3.27 |
| c18 | 7.27 | 1.17 | -0.46 | 6.37 |
| c1–c18: mean | 7.03 | 1.65 | -0.06 | 6.73 |
| standard deviation | 3.26 | 0.67 | 0.26 | 3.31 |
| 95% conf. interval | [5.40, 8.66] | [1.32, 1.98] | [-0.19, 0.07] | [5.08, 8.39] |

The statistics of the submesoscale eddy size $R$ is almost the same for anticyclones and cyclones with the mean values of 7.22 km and 7.03 km, respectively. In contrast to the eddy size $R$, the rotary characteristics of submesoscale cyclones, such as $\Omega_0$, $Dif$ and $\langle Hel \rangle$, differ considerably from respective values of the anticyclones. Namely, the ensemble mean value of $\Omega_0$ is 1.65 for cyclones and
-0.57 for anticyclones, i.e. the absolute frequency of rotation in the centre of cyclonic eddy is on average three times larger than in the anticyclone. It is also important that the cyclonic eddies are characterized by much more pronounced differential rotation (the ensemble mean value of $Dif$ is 6.73 in the cyclones versus 2.38 in the anticyclones). Lastly, there is a substantial difference in the helicity: the rotation of a particle in the mesoscale cyclonic eddy is accompanied on the average by a shift
towards the eddy centre (the ensemble mean value of $\langle Hel \rangle$ is negative (-0.06)), while in an anticyclone a particle moves on the average away from the centre (the ensemble mean value of $\langle Hel \rangle$ is positive (0.57)). It is worth noting that the 95% confidence intervals for the ensemble mean values of $Dif$ and $\langle Hel \rangle$ of the cyclonic eddies do not overlap those of the anticyclonic eddies.





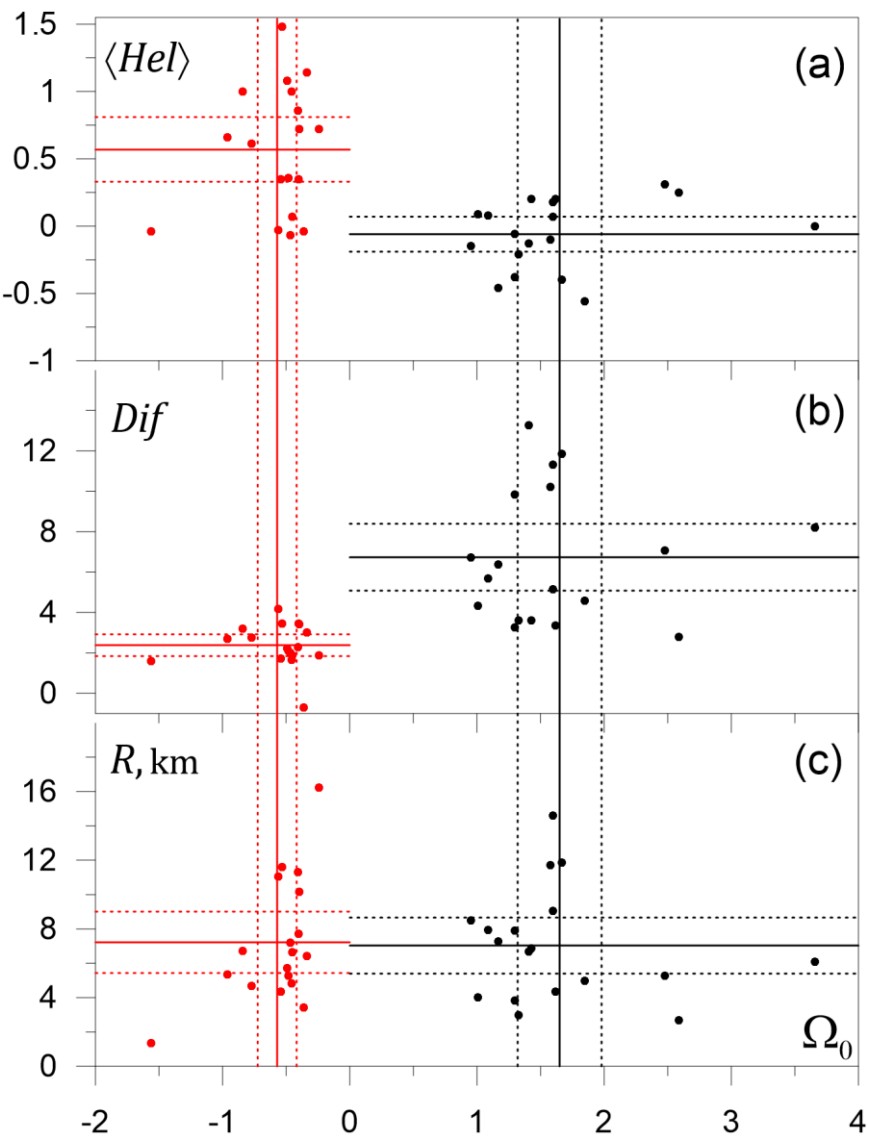

Fig. 8. Scatter plots of helicity (a) and differential rotation (b) parameters and radius (c) of a submesoscale eddy versus the normalized frequency of rotation $\Omega_0 = 2\omega_0/f$ in the eddy centre. Horizontal and vertical lines are the ensemble mean values (solid) and 95% confidence limits (dotted) of the parameters calculated separately for the anticyclonic ($\Omega_0 < 0$, red lines/symbols) and cyclonic ($\Omega_0 > 0$, black lines/symbols) eddies.





Finally, Fig. 9 presents the plots of normalized frequency of rotation $\omega/\omega_0$ versus radial distance

from the eddy centre $r/R$ of the modelled submesoscale cyclonic (a) and anticyclonic (b) eddies. The

ensemble mean curve of $\omega/\omega_0 = F(r/R)$ for cyclones/anticyclones displays much larger/smaller drop

of the rotation frequency away from the eddy centre (i.e. the more/less pronounced differential rotation)

and the positive/negative curvature (second derivative $F''$ is positive/negative).

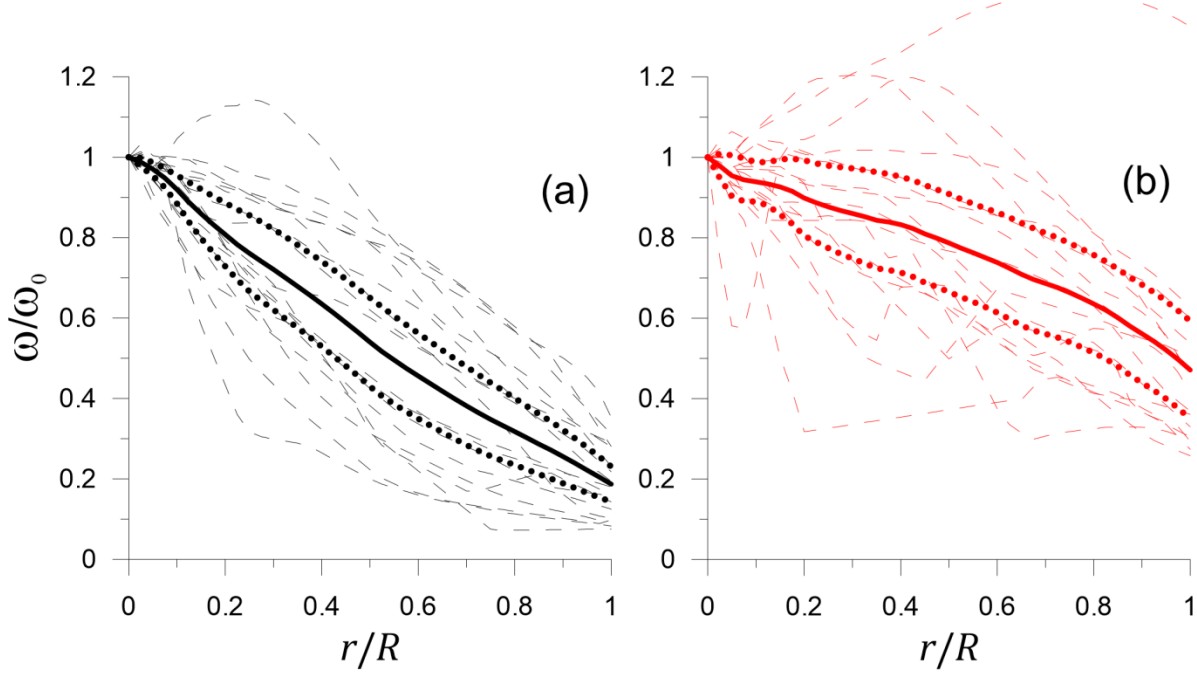


Fig. 9. Normalized dependence of angular velocity of rotation $\omega/\omega_0$ on radial distance from the eddy

centre $r/R$ in the simulated submesoscale eddies: cyclones c1–c18 (a) and anticyclones a1–a18 (b) (thin

dashed curves). The bold solid and bold dotted curves are the ensemble means and the 95% confidence

intervals, respectively. The black/red curves correspond to the cyclonic/anticyclonic eddies.

## 4    Discussion and Conclusions

As stated in the Introduction, this work is aimed to investigate the differences between rotary

characteristics of the submesoscale cyclonic and anticyclonic eddies which, in our opinion, would

explain the overwhelming dominance of cyclonic spirals on the satellite images of the sea surface





recorded in SAR, infrared and optical ranges. In this study we used numerical experiments with floating

Lagrangian particles embedded offline in a regional circulation model of the southeastern Baltic Sea with very high horizontal resolution (0.125 nautical mile grid).

The numerical experiments showed that the cyclonic spirals can be formed both from a horizontally uniform initial distribution of floating particles and from the initially lined up particle clusters during the advection time of the order of 1 day. While the formation of the predominantly cyclonic spirals from

the linear features in the course of development of horizontal shear instabilities and the mixed-layer baroclinic instabilities is a well-known effect which was thoroughly discussed by Munk et al. (2000) and Eldevik and Dysthe (2002), a quick regrouping of the floating particles from horizontally uniform distribution to cyclonic spirals in the course of advection in the submesoscale velocity field is a surprising phenomenon which was first mentioned by Väli et al. (2018).

We addressed several rotary characteristics of submesoscale eddies which could be potentially responsible for the predominant formation of cyclonic spirals such as

- normalized frequency of rotation in the eddy centre $\Omega_0 = 2\omega_0/f$ (the higher the frequency, the faster the spiral can be formed);

- differential rotation parameter $Dif = [\omega(0) - \omega(R)]/\omega(0)$ (the spirals cannot be formed from

linear features at the solid-body rotation when $Dif = 0$);

- helicity parameter $\langle Hel \rangle$ defined in Chapter 2.2 (if $\langle Hel \rangle < 0$ ($\langle Hel \rangle > 0$) the particles shift towards (away from) the eddy centre which makes the spiral more (less) visible).

To calculate $\Omega_0$, $Dif$, $\langle Hel \rangle$ and eddy radius $R$ the approach described in Chapter 2.2 was applied to the pseudo-trajectories of synthetic floating particles in a frozen velocity field (i.e. the velocity field

simulated by the circulation model for a given instant was kept stationary for the entire period of advection). As a result, we obtained estimates of $\Omega_0$, $Dif$, $\langle Hel \rangle$ and $R$ for 18 cyclonic and 18 anticyclonic submesoscale eddies simulated in the southeastern Baltic Sea in May–July 2015.

The ensemble mean value of eddy radius $R$ was 7.22 and 7.03 km for the anticyclones and cyclones, respectively, with strong overlap of the 95% confidence intervals. Therefore, one may conclude that the

submesoscale cyclonic eddies are indistinguishable by size from the submesoscale anticyclonic eddies.





In contrast to $R$, the ensemble mean values of $\Omega_0$, $Dif$ and $\langle Hel \rangle$ occurred to be substantially different for the cyclonic and anticyclonic eddies and the difference of all three rotary characteristics indicated the predominant formation of cyclonic spirals. Indeed, the ensemble mean values of $\Omega_0$, $Dif$ and $\langle Hel \rangle$ were 1.65 vs. -0.57, 6.73 vs. 2.38 and -0.06 vs. 0.57 for cyclones and anticyclones,

respectively, and the 95% confidence intervals did not overlap (see Table 1 and Fig. 8). Therefore, on the average the submesoscale cyclonic eddies, in comparison to the anticyclonic ones, rotate three times faster, have three times larger difference of the frequency of rotation between the eddy centre and the periphery, as well as display the tendency of shifting floating particles towards the eddy centre ($\langle Hel \rangle <$ 0). Note that the negative (positive) value of the helicity parameter $\langle Hel \rangle$ in the cyclonic (anticyclonic)

eddies is in accordance with the negative correlation between relative vorticity and vertical velocity in the submesoscales reported by Väli et al. (2017) (i.e. submesoscale cyclonic (anticyclonic) eddies are characterized mostly by downwelling (upwelling)).

The frequency of rotation of submesoscale eddies was found to decrease with the radial distance (i.e., the rotation is differential rather than solid-body). However, a certain similarity of solid-body rotation is

still inherent in the submesoscale anticyclones, where the difference in the frequency of rotation between the eddy centre and periphery is relatively small, and the second derivative of frequency with respect to radial distance is negative (see Fig. 9b). In contrast to the submesoscale anticyclones, in the submesoscale cyclones, where the difference in the frequency of rotation between the centre and the periphery is much larger, and the second derivative of frequency with respect to radial distance is

positive, one cannot see even a hint of the solid-body rotation (cf. Fig. 9, a and b).

## Acknowledgements

This research, including the numerical experiments with floating Lagrangian particles, was performed by Victor Zhurbas and supported in part by the Russian Foundation for Basic Research (Grant No. 18-05-80031). Germo Väli (submesoscale circulation modelling) was supported by

institutional research funding IUT 19-6 of the Estonian Ministry of Education and Research and BONUS, the joint Baltic Sea research and development programme (Art 185), funded jointly by the





European Union's Seventh Programme for research, technological development and demonstration, and by the Estonian Research Council through grant VEU17107 (BONUS INTEGRAL). An allocation of computing time on High Performance Computing cluster by the Tallinn University of Technology and by the University of Tartu is gratefully acknowledged. Natalia Kuzmina (interpretation of the numerical experiments with floating Lagrangian particles) was partially supported by budgetary financing of the Shirshov Institute of Oceanology RAS (Project No. 149-2019-0003).




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
