# Peer review of "Rotation of floating particles in submesoscale cyclonic and anticyclonic eddies: a model study for the southeastern Baltic Sea"

_Ocean Science, 2019_

## Referee Comment (RC1) · Anonymous Referee #1 · 4 Sep 2019

**Summary**

The authors use numerically derived Lagrangian trajectories and rotary statistics to understand the prevalence of distinguishable submesoscale cyclonic over anticyclonic spirals observed in tracer fields. The authors build on previous studies suggesting horizontal shear or mixed layer baroclinic instabilities wind pre-existing linear features of the tracer field that collect particles (e.g. fronts), into predominantly cyclonic spirals. The goal of this paper is to compare the rotary characteristics of cyclonic to anticyclonic spirals to provide an alternative explanation on the prevalence of cyclonic spirals in tracer fields that does not depend on the presence of linear features in the tracer field.

The authors find that both uniform and linear tracer initial conditions for Lagrangian particle releases can produce more distinct cyclonic than anticyclonic spirals in the tracer field on the order of a day, and that this is due to cyclonic spirals having larger central angular velocities, enhanced differential rotation, and negative helicity which draws material towards the center of the spiral. This is in contrast to anticyclonic spirals which take longer to develop, have smaller differential rotation, and push material away from their centers which makes them less visible in satellite imagery.

This paper is a valuable addition to existing literature, and I recommend acceptance with minor revisions. While I do not believe any new simulations etc are needed, the manuscript should expand significantly on the results and discussion sections as well as clarify several aspects of the manuscript noted below.

**General Comments**

- The abstract could use rephrasing. As written it appears that generally it's accepted that the rotary characteristics are between cyclonic and anticyclonic eddies are different, and this paper seeks to confirm that. However, in the introduction the authors do point out that their approach is different than most previous work, and this should be highlighted in the abstract. In addition the last sentence of the abstract should be written more clearly to define the three characteristics measured. The way it is worded it was hard to follow until after reading the manuscript itself. Perhaps a numbered list or commas would clarify.
- The introduction could be made stronger by including the importance of these cyclonic spirals alongside underpinning the mechanisms for their prevalence. A good deal of space is dedicated to describing their existence and previous mechanisms of formation, but not much is provided to describe the relevance. For example, what are the biological impacts given that cyanobacteria trace eddies

out so well or are there implications for eddy tracking with tracer fields? It would also be helpful to include the distinction between helicity and vorticity here, and what they tell you about a flow (e.g. for the helicity, line 170). The differential rotation parameter could also use a descriptive sentence. This will help the reader understand why you have chosen these particular aspects to compare, as well as connect to the mathematical descriptions provided later. Lastly a more thorough literature review is needed with respect to others investigating the impacts on tracer fields in anticyclonic and cyclonic eddies. For example Brannigan 2016 or Brannigan et al 2017.

- The manuscript should more clearly state that the authors are not seeking to explain the skewed tails of the vorticity distribution, only the dominance of the cyclonic spirals seen in tracer fields from satellite images. The previous studies cited provide the mechanisms favoring cyclonic spirals, and also explain why tracers highlight them over anticyclonic spirals. However in this manuscript, although Line 65 does a good job highlighting the objectives, throughout the remaining text the wording leaves it somewhat ambiguous what the authors exact intentions are with respect to both aspects of the problem. For example in the abstract and elsewhere using 'formation of spirals' is slightly misleading since the spirals are already there with respect to the velocity field. Perhaps including references to the tracer field when using this description would help clarify.
- Section 2.2 could be clarified more, specifically the rotary parameters. These should be connected to the introduction as well to give the reader intuition into the authors interpretations of them. Is  $\delta r = r_2 r_1$ ? This would help clarify the sign dependence of Helicity. How is  $\omega(0)$ , the vorticity at the center of the spiral, diagnosed here given that particles presumably tend towards stationary at the exact center? A more precise definition is needed. Does this model have the resolution to produce such results?

СЗ

- The Lagrangian particle simulations and the comparison of gridded to linearly seeded particles to understand the spiral formation should be expanded on. Can you provide justification for using surface constrained particles to understand a 3D tracer field. Do you have an idea of which mechanism for creating cyclonic spirals is most prevalent? That is submesoscale fronts are ubiquitous, what percentage of spirals tends to come from advection of particles into a strong eddy field versus reshaping of linear tracer features?
- Do you think a seasonal pattern could be isolated using these methods? For example, with an intense eddy field in winter perhaps the differences between cyclonic and anticyclonic statistics are more prominent.
- The tables should be moved to an appendix. Figure 8 should be described more thoroughly as it is the most compelling evidence in support of the hypothesis. Are the confidence intervals based on the three days of model output combined into one and are they from bootstrapping or some other method? It would be helpful to explain how these days are included. Additionally why did you choose these snapshots? Do other snapshots show similar statistics?
- The conclusion should include a paragraph at the end with a summary and the thesis of the paper reiterated.

**Specific Comments**

- What are the minimum and maximum vertical resolutions? Does this adequately resolve the helicity?
- D'Asaro 2019 might be a good reference to include as an in-situ observational compliment.

- Please fully write out dates so there is no ambiguity between Europe and the U.S. etc. For example, the way the date is presented in the caption of Figure 4 is ideal.
- Can you describe the physical intuition for the rotary characteristics of the spirals? For example why does it physically make sense that cyclonic eddies would spin faster?
- Section 2.1 Please include the method used to interpolate the topography and initial conditions etc.
- Line Number 158 Should this be |He| << 1? Why does this assumption mean you can write out the helicity with your given formula? Is this what you actually use to calculate Hel or that given in (3)?
- Line Number 220 It is not clear why this would be a validation of the model.

**Technical Comments**

- Line Number 59 "One may expect that the spirals could also be generated." Does this expectation come from observations? Please state your motivation.
- Line Number 65 Perhaps: "The objective of this work is to understand the dominance of observed cyclonic spirals by assessing differences between floating particles' rotation in submesoscale cyclonic and anticylonic spirals using high resolution modelling of the Baltic Sea."
- Line Number 71 The word 'fabulous' seems out of place here. Perhaps "The most illustrative optical images...' would work instead.
- Line Number 75 "eddies, which will be investigated..."

**C5**

- Line Number 150 Please specify that the relation is for the vertical vorticity.
- Line Number 171 Change 'It can be seen easily' to just "Large values of Dif...'
- Line Number 180 This paragraph could be worded more clearly. Specifically, ' we utilized' instead of 'we addressed'.

---

## Referee Comment (RC2) · Vladimir Ryabchenko (Referee) · 5 Sep 2019

General comments

The work examines the spiral-shaped submesoscale eddies to the southeastern Baltic Sea. The authors set out to show that the overwhelming dominance of cyclonic spirals on satellite images of the sea surface are caused by some differences between rotary characteristics of the submesoscale cyclonic and anticyclonic eddies. Using the well-known three-dimensional circulation model GETM, an evolution of eddy structures in the southeastern Baltic Sea from April 1 to October 9, 2015 is reproduced on an ultra-high resolution grid (approximately 232 m). The results are in a good agreement

with Landsat-8 true color image of the southeastern Baltic Sea on the date 07/03/2015 which shows a prominent cyclonic spiral located at a distance of about 60 km to the north-northwest from the Cape Taran. Calculated trajectories of synthetic floating Lagrangian particles embedded in the above regional circulation model showed that: 1) the cyclonic spirals are formed both from a horizontally uniform initial distribution of floating particles and from the initially lined up particles during the advection time of the order of 1 day , 2) the spirals were formed only from the linear features embedded into the submesoscale cyclonic eddies, while the linear features in the anticyclonic eddies transformed to some curves of irregular shape. Statistical processing of the trajectories of the synthetic floating particles in order to calculate the kinematic characteristics of submesoscale eddies allowed to conclude that the submesoscale cyclonic eddies differ from the anticyclonic eddies in three ways favoring the formation of the spirals: the former can be characterized by a significantly higher angular velocity and a more pronounced differential rotation as well as by a negative helicity. These features of the kinematics of submesoscale eddies were revealed for the first time, the article is important and interesting. However, I have a few questions and small comments, the answers to which I would like to receive before finally recommending the article for publication.

Specific comments

1. Studying the eddy structures and features, the authors do not refer to the surface salinity fields anywhere. At the same time, salinity is a more conservative characteristic than temperature, especially far from river estuaries, and eddy structures will probably appear clearer in salinity fields. It would be nice if the authors showed salinity fields in Fig. 4,5,6 and commented on the results.

2. Lines 96-100. The depth field in the domain of the high-resolution model (0.125 nm) has a coarser resolution (0.5 nm). I would like to hear the authors' thoughts regarding the sensitivity of the calculation results to the accuracy of the representation of the field of sea depths, especially in the coastal zone.

3. Line 119. "The high-resolution model accounts only for rivers that flow into the sea within the model domain." The meaning of the phrase is not clear. Indeed, in the high-resolution model, only rivers flowing into this area should and can be taken into account. And what else? The phrase can be deleted altogether.

4. Line 120. The procedure for obtaining the initial thermohaline fields on the coarse grid should be described in more detail. Please, indicate at least the duration of the run in which these fields were obtained.

5. In the part 2 "Material and methods", the material is not located in accordance with the order in which the results in part 3 are presented. It would be logical to isolate paragraphs Lines 179-183 and 184-185 and modify them in the new section "Synthetic floating particles approach", which is placed after section 2.1 Model setup (after line 130). In this case, the general numbering of sections will change as follows (the title of the last section was shortened): 2.1. Model setup 2.2. Synthetic floating particles approach 2.3. Rotary characteristics of submesoscale cyclones / anticyclones

6. Line 240. Why, when analyzing the results of numerical experiments in section 3.3, anticyclone marked as a17 in Fig. 6 missing?

---

## Referee Comment (RC3) · Anonymous Referee #3 · 6 Sep 2019

**Summary:**

The paper provides an interesting view on the characteristics of submesoscale cyclonic and anticyclonic eddies. By applying different rotary characteristics and a particle trajectory approach to modelled velocity fields of the southeastern Baltic Sea, the authors try to assess the differences in cyclonic and anticyclonic eddies which explain the appearance of cyclonic eddies as spirals in satellite images. The paper therefore combines a phenomenon observed in satellite images and a modelling approach to assess the submesoscale dynamics behind it.

The particle approach shows the same spirals for cyclonic eddies as observed in satellite images. The differences in the rotary characteristics of cyclonic and anticyclonic eddies can be used to motivate the formation of spirals in case of cyclonic eddies.

All in all, the paper is an interesting piece of scientific work. The applied methods could also be useful for other regions of interest in the ocean to study the occurrence of spirals that are also visible in satellite images. I recommend acceptance with minor revisions.

General Comments:

- It is not clear to me how the 18 test eddies had been chosen. Has an eddy detection tool been applied? Are they chosen by hand? Why are specifically these 18 eddies chosen? Why have only eddies in the early summer and summer been chosen, when the modelled data also cover spring and autumn? Can annual differences be expected? Does the lifetime span of the eddies impact the formation of the spirals? Are short living eddies able to develop spirals?

- Additionally, it is not clear to me if the particle trajectories are calculated only from the surface velocity field or if the three dimensional velocity field is used. If only the surface velocity field is used the question remains of how large the impact of the wind field on the surface velocity would be and what would these results show.

Specific Comments:

- I would suggest rearranging the introduction and exchanging paragraph line 57-68 with paragraph line 69-79. It seems to me more logical for the structure of the introduction: First, you talk about spirals in general (line 29-38), then about mechanisms how they could arise (line 38-50) and about the modelling of submesoscale structures (line 50-56). If you then take paragraph 69-79 and skip the sentence "As it was mentioned above, a better visualization of the cyclonic spirals is supposedly related to some differences between floating particles rotation in submesoscale cyclonic and anticyclonic eddies which will be investigated hereafter." you will give a clearer reason why to use

OSD
the Baltic Sea as a study area. Afterwards, the paragraph line 57-68 motivates and presents the objectives of the paper. To conclude the introduction, it would be helpful for the reader to give a short outline of the structure of the paper at the end of the introduction. This would make it easier for the reader to find parts in the paper that are of interest and allows the reader to skip parts they are already familiar with.

- Table 1: Is it necessary to show the whole values in the paper? A table with mean, standard deviation and 95% conf. interval for both anticyclonic and cyclonic eddies could be sufficient for the paper and much more concise. The rest of the table could be shown in the appendix or the supplementary material. Furthermore, all values are also visible in Figure 8.

- It would be helpful for the reader if ideas that has been put in brackets as in line 280ff, 309, 311 or 331ff would be outlined in full sentences without brackets to improve the reading flow.

- Discussion and conclusion: I am missing a critical reflection of the sample size of 18 eddies and the choice of the sample: Only data for one summer in one year are chosen. What about other years or seasons? The paper does not need more data yet, but open or further research questions could be mentioned in the end of the section.

Technical Comments:

- Could the definition of the eddy radius in line 160-162 also be indicated in Figure 3? It would be easier to understand the definition and why it is a valid definition for this purpose.

- Section 2.1: Model setup: What is the temporal resolution of the velocity field?

- Figure 4-7: Please indicate not only the date but also the exact time as in Figure 3.

---

## Referee Comment (RC4) · Anonymous Referee #4 · 15 Sep 2019

This is an interesting study of the reasons behind the predominance of cyclonic spirals of different substances in the marine environment of the northern hemisphere (equivalently, anticyclonic spirals in the southern hemisphere). The explanation is unexpectedly simple but clearly physically relevant. Another valuable finding is the explanation of one of the processes that systematically drives patchiness on the sea surface.

The study is sound and professionally performed. Even though Introduction provides perhaps too a massive flow of the information, it is well written and serves as a good summary to the experts from other parts of the world. The authors use an ocean model with a very high resolution that evidently is able to resolve a number of fine

(sub)mesoscale features. The simulated pattern of eddies fairly well matches the outcome of satellite remote sensing. Most likely this match partially reflects the high probability of having synoptic eddies in certain more or less fixed locations of the Baltic Sea because of the specific geometry of the sea and its shores. Even though this remark is just an observation and not critics, still I recommend making the claim on lines 220-221 a little bit weaker.

To my eyes, the use of words "linear features" (lines 57, 63 and in several occasions below) is misleading; mostly because in hydrodynamics the adjective "linear" is usually associated with properties of the underlying equations and their solutions. Thus, for many readers "linear surface features" would automatically connote "sinusoidal wave trains" even if Walter Munk used this expression in a different meaning of substances aligned into elongated patches or stripes (like mentioned on line 239).

I recommend to mention that a "sister" phenomenon of the quick regrouping of particles to cyclonic spirals (lines 302–304; Väli et al., 2018) occurs in the periphery of intense marine eddies. The associated almost explosive increase in the particle concentration in was first explored in detail in (Samuelson et al., 2012). The increase in the local concentration occurs in the rim of an anticyclonic eddy differently from that in cyclonic ones. It happens basically because of the interaction of outward motions of particles with the field of particles outside the eddy. A little bit outside of the scope of the manuscript is an attempt to quantify the associated systematic changes to the density of particles, with much lower resolution than the simulations in this manuscript, for a subbasin of the Baltic Sea (the Gulf of Finland) in terms of so-called finite-time compressibility (Kalda et al., 2014).

The entire study, in essence, signals that the well-known asymmetry of atmospheric cyclonic and anticyclonic eddies (all strong storms are cyclonic) becomes evident also in the field of ocean eddies. I guess that the reader would enjoy some comments on whether the established strong asymmetry of the rotation rates of eddies of different sign is a local property (of densely packed eddies?) or reflects a generic property of

marine eddies. This asymmetry may affect more widely the statistical parameters of surface flows (Heinloo and Toompuu, 2012) as in such occasions the average curvature of trajectories of water parcels is predominantly of one sign.

The use of English is clear and appropriate but may need at places minor corrections (e.g. on line 286 it should probably by "the radial distance" but simply "submesoscale cyclones" would do on line 292).

**Minor comments**

I recommend to be careful with the use of "rotation" of particles and to clearly distinguish rotation of particles around their own centre and (rotary) motion of particles along curved or circular trajectories. For example, the words "floating particles rotation" (line 66) could easily be misinterpreted. Similarly, "the rotation of a particle /—/ is accompanied /—/ by a shift" is ambiguous.

Some parts of the manuscript contain too long paragraphs that make it complicated to follow the line of thinking. The first paragraph of Introduction covers 27 lines that is far too much. Also, in several occasions the sentences could be split into parts for clarity.

Equation (4): it is not clear how w(0) is calculated; also there is no need for square brackets in the first expression.

Line 106: n.m. obviously stands for nautical mile but it is better to explain the abbreviation.

Line 169: perhaps it would be more exact to speak about divergence/convergence of the surface velocity field.

Line 219: use the Polish ń in Gdańsk.

The claim on line 232/233 is just a repetition of the same claim on lines 224-225.

Table 1 could be better placed in Appendix.

Line 261: "The statistics ...." contains, to my eyes, too much jargon and simply "mean" (line 267) would do the same job as "ensemble mean" (but "ensemble mean curve" on line 282 has clear meaning).

**References**

Heinloo, J., Toompuu, A., 2012. A modification of the classical Ekman model accounting for the Stokes drift and stratification effects. Environ. Fluid Mech. 12 (2), 101–113, doi: 10.1007/s10652-011-9212-5.

Kalda, J., Soomere, T., Giudici, A., 2014. On the finite-time compressibility of the surface currents in the Gulf of Finland, the Baltic Sea. J. Mar. Syst. 129, 56–65, doi: 10.1016/j.jmarsys.2012.08.010.

Samuelsen, A., Hjøllo, S.S., Johannessen, J.A., Patel, R., 2012. Particle aggregation at the edges of anticyclonic eddies and implications for distribution of biomass. Ocean Sci. 8 (3), 389–400, doi: 10.5194/os-8-389-2012.

---

## Author Comment (AC1) · 9 Oct 2019

Dear Reviewer#1,

Thank you very much for your comprehensive review of our manuscript. Please find below our replies to your comments. Note that below your comments are written in italic.

**General Comments**
• *The abstract could use rephrasing. As written it appears that generally it's accepted that the rotary characteristics are between cyclonic and anticyclonic eddies are different, and this paper seeks to confirm that. However, in the introduction the authors do point out that their approach is different than most previous work, and this should be highlighted in the abstract. In addition the last sentence of the abstract should be written more clearly to define the three characteristics measured. The way it is worded it was hard to follow until after reading the manuscript itself. Perhaps a numbered list or commas would clarify.*

    We will rephrase the abstract to highlight the novelty of the approach. A numbered list of the three characteristics assessed will be added to the last sentence.

• *The introduction could be made stronger by including the importance of these cyclonic spirals alongside underpinning the mechanisms for their prevalence. A good deal of space is dedicated to describing their existence and previous mechanisms of formation, but not much is provided to describe the relevance. For example, what are the biological impacts given that cyanobacteria trace eddies out so well or are there implications for eddy tracking with tracer fields? It would also be helpful to include the distinction between helicity and vorticity here, and what they tell you about a flow (e.g. for the helicity, line 170). The differential rotation parameter could also use a descriptive sentence. This will help the reader understand why you have chosen these particular aspects to compare, as well as connect to the mathematical descriptions provided later. Lastly a more thorough literature review is needed with respect to others investigating the impacts on tracer fields in anticyclonic and cyclonic eddies. For example Brannigan 2016 or Brannigan et al 2017.*

    We will add to the Introduction some sentences on biological/ecological impact of the spirals and provide more thorough review of recent literature on tracer fields in anticyclonic and cyclonic eddies including Brannigan (2016) and Brannigan et al. (2017). Since the helicity and differential rotation parameters are introduced later in Material and Methods chapter, we do not think it is worth to discuss them in Introduction. The distinction/relation between vorticity, horizontal divergence, helicity and differential rotation will be discussed in more detail in Material and Methods chapter.

• *The manuscript should more clearly state that the authors are not seeking to explain the skewed tails of the vorticity distribution, only the dominance of the cyclonic spirals seen in tracer fields from satellite images. The previous studies cited provide the mechanisms favoring cyclonic spirals, and also explain why tracers highlight them over anticyclonic spirals. However in this manuscript, although Line 65 does a good job highlighting the objectives, throughout the remaining text the wording leaves it somewhat ambiguous what the authors exact intentions are with respect to both aspects of the problem. For example in the abstract and elsewhere using 'formation of spirals' is slightly misleading since the spirals are already there with respect to the velocity field. Perhaps including references to the tracer field when using this description would help clarify.*

    To highlight the objectives more clearly, we will move paragraph containing Line 65 and the objectives to the end of the Introduction chapter (the last read statement is better remembered). Also we will supplement 'formation of spirals' wording with 'in the tracer field' throughout the manuscript.

• *Section 2.2 could be clarified more, specifically the rotary parameters. These should be connected to the introduction as well to give the reader intuition into the authors interpretations of them. Is $\delta = r_2 - r_1$? This would help clarify the sign dependence of Helicity. How is $\omega(0)$, the vorticity at the center of the spiral, diagnosed here given that particles presumably tend towards stationary at the exact center? A more precise definition is needed. Does this model have the resolution to produce such results?*

Right, $\delta = r_2 - r_1$, where $r_1$ and $r_2$ are the radii of two consecutive loops of a synthetic Lagragian particle, we will clearly state it in the revised manuscript. The modelled velocities were bilinearly interpolated to the current position of the particle within the grid cell. Therefore if the initial position of the particle was taken close enough to the exact centre of the eddy, the radius of the loop $r$ would be sufficiently small, e.g. smaller than the grid cell size $dx, dy = 232$ m. The frequency of particle's rotation at $r \approx 0.5dx \approx 100$ m was taken for $\omega(0)$.

To clarify the definitions of $\delta$ and $\omega(0)$ we will include the above paragraph to the revised manuscript.

• *The Lagrangian particle simulations and the comparison of gridded to linearly seeded particles to understand the spiral formation should be expanded on. Can you provide justification for using surface constrained particles to understand a 3D tracer field. Do you have an idea of which mechanism for creating cyclonic spirals is most prevalent? That is submesoscale fronts are ubiquitous, what percentage of spirals tends to come from advection of particles into a strong eddy field versus reshaping of linear tracer features?*

We realize that a scenario presented in Chapter 3.3 where the spiral in the tracer field is formed from synthetic floating particles seeded on a line passed through the centre of a mature submesoscale cyclonic or anticyclonic eddy is barely realistic because one cannot imagine a natural phenomenon that could provide such kind of seeding. However, the two other scenarios, i.e. when the spirals come from advection of uniformly seeded floating particles into velocity field of a mature eddy(see Chapter 3.2) and from reshaping of a linear tracer feature aligned to the density front in the course of development of a kind of frontal instability (the Munk's hypothesis), seem quite realistic. In our opinion, depending on the specific conditions of the ocean environment, either the first or second of two realistic scenarios may prevail.

We will add the above discussion to Discussion and Conclusions chapter.

As to the justification for using surface constrained particles to understand a 3D tracer field – we are not ready to discuss the issue which seems to be outside the scope of this study.

• *Do you think a seasonal pattern could be isolated using these methods? For example, with an intense eddy field in winter perhaps the differences between cyclonic and anticyclonic statistics are more prominent.*

We do not exclude that there is some seasonality in the differences between cyclonic and anticyclonic statistics; it deserves a separate study and right now we have nothing to add to the manuscript on the issue.

• *The tables should be moved to an appendix.*

Table 1 will be moved to Appendix.

*Figure 8 should be described more thoroughly as it is the most compelling evidence in support of the hypothesis. Are the confidence intervals based on the three days of model output combined into one and are they from bootstrapping or some other method? It would be helpful to explain how these days are included. Additionally why did you choose these snapshots? Do other snapshots show similar statistics?*

We will add an intuitive explanation of why mesoscale cyclones rotate faster than anticyclones and why helicity in cyclonic(anticyclonic) eddies is negative(positive) as follows.

"The physical intuition for faster spinning of cyclonic eddies vs anticyclonic eddies can be gained from conservation of potential vorticity in a fluid parcel (e.g., Väli et al. (2017): $(\zeta + f)\rho_z = const$, where $\rho_z$ is the vertical gradient of density. If the parcel undergoes ultimate vertical stretching ($\rho_z/\rho_z(0) \to 0$, where $\rho_z(0)$ is the initial value of $\rho_z$) given that it does not spin initially ($\zeta(0) = 0$), it will acquire unlimited cyclonic rotation: $\Omega = \zeta/f = \rho_z(0)/\rho_z - 1 \to \infty$. On the contrary, if the parcel undergoes ultimate vertical squeezing ($\rho_z/\rho_z(0) \to \infty$), it will acquire anticyclonic rotation limited from above: $\Omega \to -1 + 0$. The above considerations make it clear why in Fig. 8 in all cyclonic eddies $\Omega_0 > 1$, while in all anticyclonic eddies except one the rotation speed is within $-1 < \Omega_0 < 0$. As to the positive(negative) value of helicity in anticyclonic(cyclonic) eddy, it can be intuitively understood taking into account that the related upwelling (downwelling) implies potential energy loss and, therefore, relaxation of the eddy."

The confidence intervals are based on processing of 18 cyclonic and 18 anticyclonic eddies identified on three snapshots related to 3 days of model output; the 18 items are considered as a sample of a normally distributed quantity. The standard statistical method of assessing the confidence intervals based on the Student's t-distribution is used, and we will point it out in the revised manuscript. The three snapshots were chosen just because there were satellite images available for the dates – we will point it out in the revised manuscript too. We did not process other images because it was a time-consuming job, and we did not see any reason it would show different statistics.

• *The conclusion should include a paragraph at the end with a summary and the thesis of the paper reiterated.*

We will include such a paragraph with the summary reiterated.

***Specific Comments***
• *What are the minimum and maximum vertical resolutions? Does this adequately resolve the helicity?*

GETM belongs to the family of terrain-following models, so the vertical resolution is spatially varying. In the shallow region, the cell thickness in the surface layer was less than 0.5 m. In the study area (Gdansk Bay, SW Baltic Proper) the uppermost cell thickness did not exceed 1.8 m and could be regarded as relatively high resolution. Therefore, the authors feel that the helicity was adequately resolved by the model. Also we see no reason to argue that spatially varying vertical resolution could bias estimates of helicity.

In view of this remark, we will add info on the vertical resolution in the surface layer to the revised manuscript.

• *D'Asaro 2019 might be a good reference to include as an in-situ observational compliment.*

We will add D'Asaro et al. (2018) to the References and a short description of their main finding to the Introduction.

• *Please fully write out dates so there is no ambiguity between Europe and the U.S. etc. For example, the way the date is presented in the caption of Figure 4 is ideal.*

Everywhere in the manuscript we will adhere to the European/British style of writing dates: dd-mm-yy, for example, 15 May 2015.

*• Can you describe the physical intuition for the rotary characteristics of the spirals? For example why does it physically make sense that cyclonic eddies would spin faster?*

The physical intuition for faster spinning of cyclonic eddies vs anticyclonic eddies can be gained from conservation of potential vorticity in a fluid parcel: $(\zeta + f)\rho_z = const$, where $\rho_z$ is the vertical gradient of density. If the parcel undergoes ultimate vertical stretching ($\rho_z/\rho_z(0) \to 0$, where $\rho_z(0)$ is the initial value of $\rho_z$) given that it does not spin initially ($\zeta(0) = 0$), it will acquire unlimited cyclonic rotation: $\Omega = \zeta/f = \rho_z(0)/\rho_z - 1 \to \infty$. On the contrary, if the parcel undergoes ultimate vertical squeezing ($\rho_z/\rho_z(0) \to \infty$), it will acquire anticyclonic rotation limited from above: $\Omega \to -1 + 0$. The above considerations make it clear why in Fig. 8 in all cyclonic eddies $\Omega_0 > 1$, while in all anticyclonic eddies except one the rotation speed is within $-1 < \Omega_0 < 0$. As to the positive(negative) value of helicity in anticyclonic(cyclonic) eddy, it can be intuitively understood taking into account that the related upwelling (downwelling) implies potential energy loss and, therefore, relaxation of the eddy.

The above reasoning will be included to the Discussion and Conclusions chapter.

*• **Section 2.1** - Please include the method used to interpolate the topography and initial conditions etc.*

We will rewrite this section as follows.

Previous text on line 99: "The digital topography of the Baltic Sea with the resolution of 0.5 nautical miles was obtained from the Baltic Sea Bathymetry Database (http://data.bshc.pro/) and interpolated to the resolution required."

will be replaced with:

"The digital topography of the Baltic Sea with the resolution of 0.25 nautical miles (approximately 500 m) was obtained from the Baltic Sea Bathymetry Database (http://data.bshc.pro/) and interpolated bi-linearly to approximately 250 m resolution."

Please notice, that there was also a mistake in the original text, for which the authors apologize.

Previous text on line 104: "For the open boundary conditions the one-way nesting approach is used and the results from the coarse resolution model are utilized at the boundaries."

will be replaced with:

"For the open boundary conditions the one-way nesting approach was used and the results from the coarse resolution model were utilized at the boundaries. Sea-level fluctuations with 1-hourly resolution and temperature, salinity and current velocity profiles with 3-hourly resolution were interpolated using the nearest neighbor method in space to the higher resolution grid. In addition, the profiles were extended to the bottom of the high resolution model."

Previous text on line 120: "The initial thermohaline field was obtained from the coarse resolution model for 1 April 2015 and interpolated to the high-resolution model grid."

will be replaced with:

"The initial thermohaline field was obtained from the coarse resolution model for 1 April 2015 and interpolated using the nearest neighbor method to the high-resolution model grid. In addition, as the adaptive vertical coordinates were used in both setups, the T/S profiles from coarse resolution were linearly interpolated to fixed 10 m vertical resolution before interpolation to the high resolution."

• *Line Number 158 - Should this be* $Hel \ll 1$*? Why does this assumption mean you can write out the helicity with your given formula? Is this what you actually use to calculate Hel or that given in (3)?*

There was an inaccuracy in Lines 168-169 for which the authors apologize. In the revised manuscript we will replace

"If $Hel \ll 1$ in an axisymmetric eddy, it can be presented as $Hel = 2\pi V_r/V_\varphi$, where $V_r$ is the radial component of velocity."

with

"In the case of the axisymmetric eddy the helicity parameter (3) can be rewritten as $Hel = 2\pi V_r/V_\varphi$, where $V_r$ is the radial component of velocity, and in the case of no differential rotation/divergence in the axisymmetric eddy it can be expressed through the ratio of divergence $D = 2V_r/r = const$ and vorticity $\zeta = 2V_\varphi/r = const$ as $Hel = 2\pi D/\zeta$. In view of continuity the vertical velocity $W$, which is responsible for upwelling/downwelling in the eddy, is determined near the surface by horizontal divergence $D$ and depth $z$ as $W = zD$."

Actually we used (3) and not $Hel = 2\pi V_r/V_\varphi$ to calculate *Hel* (because real/simulated eddies are not exactly axisymmetric.)

• *Line Number 220 - It is not clear why this would be a validation of the model.*

In view of this remark the previous text on line 220:

"The possibility to identify the observed vortex pair in the simulated fields can be considered as a validation of the model."

will be replaced with

"The fact that a vortex pair of almost the same size and orientation was modeled in almost the same place and at the same time as the observed vortex pair can be considered as a validation of the model."

***Technical Comments***
• *Line Number 59 - "One may expect that the spirals could also be generated." Does this expectation come from observations? Please state your motivation.*

We will add to here a reference to numerical experiments on floating particles advection (Väli et al., 2018).

• *Line Number 65 - Perhaps: "The objective of this work is to understand the dominance of observed cyclonic spirals by assessing differences between floating particles' rotation in submesoscale cyclonic and anticylonic spirals using high resolution modelling of the Baltic Sea."*

Thanks a lot! We will replace the text on Line 65 with the above sentence.

• *Line Number 71 - The word 'fabulous' seems out of place here. Perhaps "The most illustrative optical images...' would work instead.*

Ok, we will change "Fabulous" for "The most illustrative"

• *Line Number 75 - "eddies, which will be investigated..."*
Line 75 says

"vortex pair consisting of coupled cyclonic and anticyclonic eddies, the latter located at about 30 km to".

We do not understand the sense of using "eddies, which will be investigated..." to here.

• *Line Number 150 - Please specify that the relation is for the vertical vorticity.*

We will change "vorticity" for "vertical vorticity".

• *Line Number 171 - Change 'It can be seen easily' to just "Large values of Dif...'*

"It can be easily seen that the" will be dropped.

• *Line Number 180 - This paragraph could be worded more clearly. Specifically, ' we utilized' instead of 'we addressed'.*

We will replace
"Apart from the above defined rotary characteristics of submesoscale eddies calculated from frozen velocity field, we addressed some numerical experiments with the deployment of synthetic floating particles in the modelled non-stationary (not frozen) velocity field, namely, when initially the particles were uniformly distributed on the sea surface, and when initially the particles formed a linear feature (i.e. a line) passing through the centre of a cyclonic or anticyclonic eddy"

with

"Apart from the above defined rotary characteristics of submesoscale eddies calculated from frozen velocity field, we utilized some numerical experiments with the deployment of synthetic floating particles in the modelled non-stationary (not frozen) velocity field, namely, when the particles were uniformly seeded on the sea surface, and when the particles were seeded on a line passed through the centre of a cyclonic or anticyclonic eddy."

---

## Author Comment (AC3) · 16 Oct 2019

Dear Reviewer #3,

Thank you very much for your comprehensive review of our manuscript. Please find below our replies to your comments. Note that below your comments are written in italic.

*General Comments:*
*- It is not clear to me how the 18 test eddies had been chosen. Has an eddy detection tool been applied? Are they chosen by hand? Why are specifically these 18 eddies chosen? Why have only eddies in the early summer and summer been chosen when the modelled data also cover spring and autumn? Can annual differences be expected? Does the lifetime span of the eddies impact the formation of the spirals? Are short living eddies able to develop spirals?*

The snapshots of 15 May, 8 June and 3 July 2015 were chosen for the analysis of submesoscale eddy field just because they corresponded to three days in the beginning of the modelling period for which there were satellite images available (one of the images is presented in Fig. 1). The number of vortices to be processed (18 cyclones and 18 anticyclones) was determined by a compromise between the desire to obtain reliable statistics and not spend too much time on it (the procedure for calculating the rotary characteristics of the eddy described in Chapter 2.2 was not fully automated and therefore quite time-consuming). We will point the above circumstances out in the revised manuscript.

We do not exclude that there is some seasonality in the rotary characteristics of submesoscale eddies as well as some dependence on the eddy age and lifespan. These issues could not be investigated in the framework of this article and we would indicate them as a possible direction for future research in the end of revised manuscript.

*- Additionally, it is not clear to me if the particle trajectories are calculated only from the surface velocity field or if the three dimensional velocity field is used. If only the surface velocity field is used the question remains of how large the impact of the wind field on the surface velocity would be and what would these results show.*

The particles were advected by velocities simulated in the uppermost sigma-layer whose thickness did not exceed 1.8 m – we will point it out in the revised manuscript. We agree with the reviewer that the velocity field used has a component directly caused by wind stress (i.e. the Ekman wind drift), but this component is unlikely to bias the rotary characteristics of submesoscale eddies in view of huge difference in horizontal length scale between atmospheric cyclones / anticyclones (~1000 km) and submesocale eddies in the ocean (~10 km).

Specific Comments:
*- I would suggest rearranging the introduction and exchanging paragraph line 57-68 with paragraph line 69-79. It seems to me more logical for the structure of the introduction: First, you talk about spirals in general (line 29-38), then about mechanisms how they could arise (line 38-50) and about the modelling of submesoscale structures (line 50-56). If you then take paragraph 69-79 and skip the sentence "As it was mentioned above, a better visualization of the cyclonic spirals is supposedly related to some differences between floating particles rotation in submesoscale cyclonic and anticyclonic eddies which will be investigated hereafter." you will give a clearer reason why to use the Baltic Sea as a study area. Afterwards, the paragraph line 57-68 motivates and presents the objectives of the paper. To conclude the introduction, it would be helpful for the reader to give a short outline of the structure of the paper at the end of the introduction. This would make it easier for the reader to find parts in the paper that are of interest and allows the reader to skip parts they are already familiar with.*

We agree with this remark.

Proposed action: Paragraph line 57-68 will be moved to the end of Introduction.

*- Table 1: Is it necessary to show the whole values in the paper? A table with mean, standard deviation and 95% conf. interval for both anticyclonic and cyclonic eddies could be sufficient for the paper and much more concise. The rest of the table could be shown in the appendix or the supplementary material. Furthermore, all values are also visible in Figure 8.*

Table 1 will be moved to Appendix.

*- It would be helpful for the reader if ideas that has been put in brackets as in line 280ff, 309, 311 or 331ff would be outlined in full sentences without brackets to improve the reading flow.*

We will avoid using brackets in sentences like that of 280 and 309. However, in our opinion, a scientific article differs from fiction in that it is more difficult to read and the reader has to be prepared for this.

*- Discussion and conclusion: I am missing a critical reflection of the sample size of 18 eddies and the choice of the sample: Only data for one summer in one year are chosen. What about other years or seasons? The paper does not need more data yet, but open or further research question could be mentioned in the end of the section.*

A reflection on the sample size and the choice of the sample will be added to Chapter 2.2 (see above our reply to General Comments).

The differences in rotary characteristics of submesoscale cyclonic and anticyclonic eddies were statistically assessed from a limited model output for early summer 2015 in the southeast Baltic Sea, and we could not exclude seasonal and interannual variability of the studied parameters as well as some dependences on the eddy age and lifespan. These issues could be the subject for future research.

The latter paragraph will be added to the end of Discussion & Conclusion chapter.

*Technical Comments:*
*- Could the definition of the eddy radius in line 160-162 also be indicated in Figure 3? It would be easier to understand the definition and why it is a valid definition for this purpose.*

Fig. 3 was designed to explain the definitions of $\omega$ and $r$, the radius and frequency of the particle rotation, and it was not intended / suited to explain the definition of the eddy radius $R$. Meantime, to our mind, verbal definition of $R$ in line 160-162 seems quite clear, unambiguous and constructive: "If a particle is deployed at a large enough distance from the eddy centre, the pseudo-trajectory will inevitably cease to be looped, and the largest $r$ calculated from a still loop-shaped trajectory is taken for eddy radius $R$".

Proposed action: none.

*- Section 2.1: Model setup: What is the temporal resolution of the velocity field?*

The temporal resolution of the velocity field was 10 minutes – the output from the model has been saved with 10 minute resolution for further numerical calculations of particle trajectories. We referred the reader to Väli et al. (2018) for details.

Proposed action: we will indicate the temporal resolution in the end of Chapter 2.2.

*- Figure 4-7: Please indicate not only the date but also the exact time as in Figure 3.*
We will indicate exact time in the captions of Figs. 4-7.

---

## Author Response (AR1)

Dear Dr. Delhez,

We took into account all remarks/comments of four reviewers you had appointed to evaluate our manuscript. Hereafter please find our responses to the reviews.

Kind regards,

Victor Zhurbas, Germo Väli, and Natalia Kuzmina

**Review #1**

Dear Reviewer#1,

Thank you very much for your comprehensive review of our manuscript. Please find below our replies to your comments. Note that below your comments are written in italic.

***General Comments***
• *The abstract could use rephrasing. As written it appears that generally it's accepted that the rotary characteristics are between cyclonic and anticyclonic eddies are different, and this paper seeks to confirm that. However, in the introduction the authors do point out that their approach is different than most previous work, and this should be highlighted in the abstract. In addition the last sentence of the abstract should be written more clearly to define the three characteristics measured. The way it is worded it was hard to follow until after reading the manuscript itself. Perhaps a numbered list or commas would clarify.*

We rephrased the abstract to highlight the novelty of the approach. A numbered list of the three characteristics assessed was added to the last sentence (Lines 21–26).

• *The introduction could be made stronger by including the importance of these cyclonic spirals alongside underpinning the mechanisms for their prevalence. A good deal of space is dedicated to describing their existence and previous mechanisms of formation, but not much is provided to describe the relevance. For example, what are the biological impacts given that cyanobacteria trace eddies out so well or are there implications for eddy tracking with tracer fields? It would also be helpful to include the distinction between helicity and vorticity here, and what they tell you about a flow (e.g. for the helicity, line 170). The differential rotation parameter could also use a descriptive sentence. This will help the reader understand why you have chosen these particular aspects to compare, as well as connect to the mathematical descriptions provided later. Lastly a more thorough literature review is needed with respect to others investigating the impacts on tracer fields in anticyclonic and cyclonic eddies. For example Brannigan 2016 or Brannigan et al 2017.*

We added to the Introduction some sentences on biological/ecological impact of the spirals (Lines 68–70, 78–80, 87–92) and provide more thorough review of recent literature on tracer fields in anticyclonic and cyclonic eddies including Brannigan (2016) and Brannigan et al. (2017) (Lines 64–82). Since the helicity and differential rotation parameters are introduced later in Material and Methods chapter, we do not think it is worth to discuss them in Introduction. The distinction/relation between vorticity, horizontal divergence and helicity as well as the definition of the differential rotation parameter were discussed in more detail in Material and Methods chapter (Lines 200–206, 219–221).

• *The manuscript should more clearly state that the authors are not seeking to explain the skewed tails of the vorticity distribution, only the dominance of the cyclonic spirals seen in*

*tracer fields from satellite images. The previous studies cited provide the mechanisms favoring cyclonic spirals, and also explain why tracers highlight them over anticyclonic spirals. However in this manuscript, although Line 65 does a good job highlighting the objectives, throughout the remaining text the wording leaves it somewhat ambiguous what the authors exact intentions are with respect to both aspects of the problem. For example in the abstract and elsewhere using 'formation of spirals' is slightly misleading since the spirals are already there with respect to the velocity field. Perhaps including references to the tracer field when using this description would help clarify.*

To highlight the objectives more clearly, we moved paragraph containing Line 65 and the objectives to the end of the Introduction chapter (Lines 103–116; the last read statement is better remembered). Also we supplemented 'formation of spirals' wording with 'in the tracer field' throughout the manuscript.

*• Section 2.2 could be clarified more, specifically the rotary parameters. These should be connected to the introduction as well to give the reader intuition into the authors interpretations of them. Is _$\delta = r_2 - r_1$? This would help clarify the sign dependence of Helicity. How is $\omega(0)$, the vorticity at the center of the spiral, diagnosed here given that particles presumably tend towards stationary at the exact center? A more precise definition is needed. Does this model have the resolution to produce such results?*

Right, $\delta = r_2 - r_1$, where $r_1$ and $r_2$ are the radii of two consecutive loops of a synthetic Lagrangian particle, we clearly stated it in the revised manuscript (Lines199–202). The modelled velocities were bilinearly interpolated to the current position of the particle within the grid cell. Therefore if the initial position of the particle was taken close enough to the exact centre of the eddy, the radius of the loop $r$ would be sufficiently small, e.g. smaller than the grid cell size $dx, dy = 232$ m. The frequency of particle's rotation at $r \approx 0.5 dx \approx 100$ m was taken for $\omega(0)$.

To clarify the definitions of $\delta$ and $\omega(0)$ we included the above paragraph to the revised manuscript (see Lines 199-214).

*• The Lagrangian particle simulations and the comparison of gridded to linearly seeded particles to understand the spiral formation should be expanded on. Can you provide justification for using surface constrained particles to understand a 3D tracer field. Do you have an idea of which mechanism for creating cyclonic spirals is most prevalent? That is submesoscale fronts are ubiquitous, what percentage of spirals tends to come from advection of particles into a strong eddy field versus reshaping of linear tracer features?*

We realize that a scenario presented in Chapter 3.3 where the spiral in the tracer field is formed from synthetic floating particles seeded on a line passed through the centre of a mature submesoscale cyclonic or anticyclonic eddy is barely realistic because one can hardly imagine a natural phenomenon capable to provide such kind of seeding. However, the two other scenarios, i.e. when the spirals come from advection of uniformly seeded floating particles into velocity field of a mature eddy(see Chapter 3.2) and from reshaping of a linear tracer feature aligned to the density front in the course of development of a kind of frontal instability (the Munk's hypothesis), seem quite realistic. In our opinion, depending on the specific conditions of the ocean environment, either the first or second of two realistic scenarios may prevail.

We added the above paragraph to Discussion and Conclusions chapter (Lines 450–459).

As to the justification for using surface constrained particles to understand a 3D tracer field – some discussion on the issue motivated by Brannigan (2016) findings can be found on Lines 64–68, 266–269.

*• Do you think a seasonal pattern could be isolated using these methods? For example, with an intense eddy field in winter perhaps the differences between cyclonic and anticyclonic statistics are more prominent.*

We do not exclude that there is some seasonality in the differences between cyclonic and anticyclonic statistics; it deserves a separate study and we have pointed it out in the end of Discussion and Conclusions chapter (Lines 464–468).

*• The tables should be moved to an appendix.*

Table 1 was moved to Appendix.

*Figure 8 should be described more thoroughly as it is the most compelling evidence in support of the hypothesis. Are the confidence intervals based on the three days of model output combined into one and are they from bootstrapping or some other method? It would be helpful to explain how these days are included. Additionally why did you choose these snapshots? Do other snapshots show similar statistics?*

The confidence intervals are based on processing of 18 cyclonic and 18 anticyclonic eddies identified on three snapshots related to 3 days of model output; the 18 items are considered as a sample of a normally distributed quantity. The standard statistical method of assessing the confidence intervals based on the Student's t-distribution is used, and we pointed it out in the revised manuscript. The three snapshots were chosen just because there were satellite images available for the dates – we pointed it out in the revised manuscript too. We did not process other images because it was a time-consuming job, and we did not see any reason it would show different statistics (see Lines 251–253, 273–278, 345–348).

*• The conclusion should include a paragraph at the end with a summary and the thesis of the paper reiterated.*

We included such a paragraph with the summary reiterated (Lines 460–468).

**Specific Comments**
*• What are the minimum and maximum vertical resolutions? Does this adequately resolve the helicity?*

GETM belongs to the family of terrain-following models, so the vertical resolution is spatially varying. In the shallow region, the cell thickness in the surface layer was less than 0.5 m. In the study area (Gdansk Bay, SW Baltic Proper) the uppermost cell thickness did not exceed 1.8 m and could be regarded as relatively high resolution. Therefore, the authors feel that the helicity was adequately resolved by the model. Also we see no reason to argue that spatially varying vertical resolution could bias estimates of helicity.

In view of this remark, we added info on the vertical resolution in the surface layer to the revised manuscript Lines 130–133).

*• D'Asaro 2019 might be a good reference to include as an in-situ observational compliment.*

We added D'Asaro et al. (2018) to the References and a short description of their main finding to the Introduction (Lines 70–76).

*• Please fully write out dates so there is no ambiguity between Europe and the U.S. etc. For example, the way the date is presented in the caption of Figure 4 is ideal.*

Everywhere in the manuscript we adhered to the European/British style of writing dates: dd-mm-yy, for example, 15 May 2015.

• *Can you describe the physical intuition for the rotary characteristics of the spirals? For example why does it physically make sense that cyclonic eddies would spin faster?*

The physical intuition for faster spinning of cyclonic eddies vs anticyclonic eddies can be gained from conservation of potential vorticity in a fluid parcel: $(\zeta + f)\rho_z = const$, where $\rho_z$ is the vertical gradient of density. If the parcel undergoes ultimate vertical stretching ($\rho_z/\rho_z(0) \to 0$, where $\rho_z(0)$ is the initial value of $\rho_z$) given that it does not spin initially ($\zeta(0) = 0$), it will acquire unlimited cyclonic rotation: $\Omega = \zeta/f = \rho_z(0)/\rho_z - 1 \to \infty$. On the contrary, if the parcel undergoes ultimate vertical squeezing ($\rho_z/\rho_z(0) \to \infty$), it will acquire anticyclonic rotation limited from above: $\Omega \to -1 + 0$. The above considerations make it clear why in Fig. 8 in all cyclonic eddies $\Omega_0 > 1$, while in all anticyclonic eddies except one the rotation speed is within $-1 < \Omega_0 < 0$. As to the positive(negative) value of helicity in anticyclonic(cyclonic) eddy, it can be intuitively understood taking into account that the related upwelling (downwelling) implies potential energy loss and, therefore, relaxation of the eddy.

The above reasoning was included to the Discussion and Conclusions chapter (Lines 429–440).

• **Section 2.1** - *Please include the method used to interpolate the topography and initial conditions etc.*

We rewrote this section as follows.

Previous text on line 99: "The digital topography of the Baltic Sea with the resolution of 0.5 nautical miles was obtained from the Baltic Sea Bathymetry Database (http://data.bshc.pro/) and interpolated to the resolution required."

was replaced with:

"The digital topography of the Baltic Sea with the resolution of 500 m (approximately 0.25 nautical miles) was obtained from the Baltic Sea Bathymetry Database (http://data.bshc.pro/) and interpolated bi-linearly to approximately 250 m resolution."

Please notice that there was also a mistake in the original text, for which the authors apologize.

Previous text on line 104: "For the open boundary conditions the one-way nesting approach is used and the results from the coarse resolution model are utilized at the boundaries."

was replaced with:

"For the open boundary conditions the one-way nesting approach was used and the results from the coarse resolution model were utilized at the boundaries. Sea-level fluctuations with 1-hourly resolution and temperature, salinity and current velocity profiles with 3-hourly resolution were interpolated using the nearest neighbor method in space to the higher resolution grid. In addition, the profiles were extended to the bottom of the high resolution model."

Previous text on line 120: "The initial thermohaline field was obtained from the coarse resolution model for 1 April 2015 and interpolated to the high-resolution model grid."

was replaced with:

"The initial thermohaline field was obtained from the coarse resolution model for 1 April 2015 and interpolated using the nearest neighbour method to the high-resolution model grid. In addition, as the adaptive vertical coordinates were used in both setups, the T/S profiles from coarse resolution were linearly interpolated to fixed 10 m vertical resolution before interpolation to the high resolution."

*• Line Number 158 - Should this be Hel ≪ 1? Why does this assumption mean you can write out the helicity with your given formula? Is this what you actually use to calculate Hel or that given in (3)?*

There was an inaccuracy in Lines 168-169 for which the authors apologize. In the revised manuscript we replaced

"If $Hel \ll 1$ in an axisymmetric eddy, it can be presented as $Hel = 2\pi V_r/V_\varphi$, where $V_r$ is the radial component of velocity."

with

"In the case of the axisymmetric eddy the helicity parameter (3) can be rewritten as $Hel = 2\pi V_r/V_\varphi$, where $V_r$ is the radial component of velocity, and in the case of no differential rotation/divergence in the axisymmetric eddy it can be expressed through the ratio of divergence $D = 2V_r/r = const$ and vorticity $\zeta = 2V_\varphi/r = const$ as $Hel = 2\pi D/\zeta$. In view of continuity the vertical velocity $W$, which is responsible for upwelling/downwelling in the eddy, is determined near the surface by horizontal divergence $D$ and depth $z$ as $W = zD$."

Actually we used (3) and not $Hel = 2\pi V_r/V_\varphi$ to calculate $Hel$ (because real/simulated eddies are not exactly axisymmetric.)

*• Line Number 220 - It is not clear why this would be a validation of the model.*

In view of this remark the previous text on line 220:

"The possibility to identify the observed vortex pair in the simulated fields can be considered as a validation of the model."

was replaced with

"The fact that a vortex pair of almost the same size and orientation was modeled in almost the same place and at the same time as the observed vortex pair can be considered as a validation of the model."

***Technical Comments***
*• **Line Number 59** - "One may expect that the spirals could also be generated." Does this expectation come from observations? Please state your motivation.*

We added to here a reference to numerical experiments on floating particles advection (Väli et al., 2018).

*• **Line Number 65** - Perhaps: "The objective of this work is to understand the dominance of observed cyclonic spirals by assessing differences between floating particles' rotation in submesoscale cyclonic and anticylonic spirals using high resolution modelling of the Baltic Sea."*

Thanks a lot! We replaced the text on Line 65 with the above sentence (see Lines 113–116).

*• **Line Number 71** - The word 'fabulous' seems out of place here. Perhaps "The most illustrative optical images...' would work instead.*

We changed "Fabulous" for "The most illustrative"

*• **Line Number 75** - "eddies, which will be investigated..."*

Line 75 of the previous manuscript said

"vortex pair consisting of coupled cyclonic and anticyclonic eddies, the latter located at about 30 km to".

We do not understand the sense of using "eddies, which will be investigated..." to here.

• *Line Number 150 - Please specify that the relation is for the vertical vorticity.*

We changed "vorticity" for "vertical vorticity".

• *Line Number 171 - Change 'It can be seen easily' to just "Large values of Dif...'*

"It can be easily seen that the" was dropped.

• *Line Number 180 - This paragraph could be worded more clearly. Specifically, ' we utilized' instead of 'we addressed'.*

We replaced

"Apart from the above defined rotary characteristics of submesoscale eddies calculated from frozen velocity field, we addressed some numerical experiments with the deployment of synthetic floating particles in the modelled non-stationary (not frozen) velocity field, namely, when initially the particles were uniformly distributed on the sea surface, and when initially the particles formed a linear feature (i.e. a line) passing through the centre of a cyclonic or anticyclonic eddy"

with

"Apart from the above defined rotary characteristics of submesoscale eddies calculated from frozen velocity field, we utilized some numerical experiments with the deployment of synthetic floating particles in the modelled non-stationary (not frozen) velocity field, namely, when the particles were uniformly seeded on the sea surface, and when the particles were seeded on a line passed through the centre of a cyclonic or anticyclonic eddy."

**Review #2**

Dear Dr. Vladimir Ryabchenko,

Thank you very much for your comprehensive review of our manuscript. Please find below our replies to your comments. Note that below your comments are written in italic.

*General comments*
*... However, I have a few questions and small comments, the answers to which I would like to receive before finally recommending the article for publication.*
*Specific comments*
*1. Studying the eddy structures and features, the authors do not refer to the surface salinity fields anywhere. At the same time, salinity is a more conservative characteristic than temperature, especially far from river estuaries, and eddy structures will probably appear clearer in salinity fields. It would be nice if the authors showed salinity fields in Fig. 4,5,6 and commented on the results.*

Fig. 6 supplemented by salinity field is given below.

[Figure]

Fig. 6 supplemented by salinity field.

Despite the salinity is believed to be a more conservative tracer than temperature, the spirals in the temperature field seem more pronounced to those in the salinity field. Probably, the reason lies in the fact that the mixed layer under the conditions of the seasonal thermocline is characterized by small but noticeable vertical temperature gradients and vanishingly small vertical salinity gradients. Following Branningan (2016), it can be assumed that the spirals in the surface temperature field are associated with the alternation of upwelling/downelling cells with transverse wave length of the order of 1 km in the mixed layer of a differentially rotating eddy, caused by submesoscale instabilities.

In view of this remark, we supplemented Figs. 4-6 with the salinity panels and added to the revised manuscript the above comment/paragraph (Lines 262–269).

*2. Lines 96-100. The depth field in the domain of the high-resolution model (0.125 nm) has a coarser resolution (0.5 nm). I would like to hear the authors' thoughts regarding the sensitivity of the calculation results to the accuracy of the representation of the field of sea depths, especially in the coastal zone.*

The authors apologize for mistake in the manuscript – the BSBD data has original resolution 500 m and not 0.5 n.m, so the resolution of the data was better than what reviewer might have assumed based on the original text (see Lines 133–135).

Regarding the sensitivity of the calculation results to the representation of the field of sea depths, the authors have the feeling it does not really matter if the original bathymetry had also been on 0.125 n.m resolution. As the GETM is so-called sigma-layer family model, which has the number of layers constant over the computational grid in contrast to the z-coordinate models,

some smoothing of the bathymetry is required to reduce the possibility for pressure gradient errors and also to make it more stable numerically. Therefore, we also applied a weak smoothing for the topography and in the end, the impact of the resolution of the original bathymetry was reduced.

*3. Line 119. "The high-resolution model accounts only for rivers that flow into the sea within the model domain." The meaning of the phrase is not clear. Indeed, in the high-resolution model, only rivers flowing into this area should and can be taken into account. And what else? The phrase can be deleted altogether.*

Indeed, only those rivers that are within the model domain, can be included. The sentence meaning was that we took into account all the rivers also in the high-resolution model even for the short period. For instance Laanemets et al (2011) only used the river Neva in their model simulations. In any case, we removed the sentence.

*4. Line 120. The procedure for obtaining the initial thermohaline fields on the coarse grid should be described in more detail. Please, indicate at least the duration of the run in which these fields were obtained.*

Indeed, we have not stated the initial conditions for the thermohaline fields of the coarse resolution model in the manuscript. They were obtained from the Copernicus Marine Service using the reanalysis product for 1989–2015. The corresponding text regarding the model setup was improved in the revised manuscript (see Lines 145–148, 164–168).

*5. In the part 2 "Material and methods", the material is not located in accordance with the order in which the results in part 3 are presented. It would be logical to isolate paragraphs Lines 179-183 and 184-185 and modify them in the new section "Synthetic floating particles approach", which is placed after section 2.1 Model setup (after line 130). In this case, the general numbering of sections will change as follows (the title of the last section was shortened): 2.1. Model setup 2.2. Synthetic floating particles approach 2.3. Rotary characteristics of submesoscale cyclones / anticyclones.*

In our opinion, the phrase "Synthetic floating particles approach" includes a wide range of problems that is outside the scope of this study, and therefore its use as the title of a subchapter of the manuscript does not seem appropriate (seems too generalized). We would prefer the old, more specific title "Application of synthetic floating particles approach to extract rotary characteristics of submesoscale cyclones/anticyclones", which athough a bit long, but fully consistent with the content of the chapter.

*6. Line 240. Why, when analyzing the results of numerical experiments in section 3.3, anticyclone marked as a17 in Fig. 6 missing?*

The anticyclone a17 was omitted because this eddy occurred to be too young: it could not be clearly identified two days before 3 July 2015 to seed synthetic particles on a line passed through its centre an therefore provide a numerical experiment on advection of the particles.

We included the above explanation to the revised manuscript (Lines 319–321).

**Review #3**

Dear Reviewer #3,

Thank you very much for your comprehensive review of our manuscript. Please find below our replies to your comments. Note that below your comments are written in italic.

*General Comments:*
*- It is not clear to me how the 18 test eddies had been chosen. Has an eddy detection tool been applied? Are they chosen by hand? Why are specifically these 18 eddies chosen? Why have only eddies in the early summer and summer been chosen when the modelled data also cover spring and autumn? Can annual differences be expected? Does the lifetime span of the eddies impact the formation of the spirals? Are short living eddies able to develop spirals?*

The snapshots of 15 May, 8 June and 3 July 2015 were chosen for the analysis of submesoscale eddy field just because they corresponded to three days in the beginning of the modelling period for which there were satellite images available (one of the images is presented in Fig. 1). The number of vortices to be processed (18 cyclones and 18 anticyclones) was selected as a compromise between the requirement to provide statistically significant results and the time spent on obtaining a suitable sample of eddies. Note that the procedure for calculating the rotary characteristics of the eddy described in Chapter 2.2 was not fully automated and, therefore, was quite time-consuming. We pointed the above circumstances out in the revised manuscript (Lines 251–253, 273–278).

We do not exclude that there is some seasonality in the rotary characteristics of submesoscale eddies as well as some dependence on the eddy age and lifespan. These issues could not be investigated in the framework of this article and we indicated them as a possible direction for future research in the end of revised manuscript (Lines 464–468).

*- Additionally, it is not clear to me if the particle trajectories are calculated only from the surface velocity field or if the three dimensional velocity field is used. If only the surface velocity field is used the question remains of how large the impact of the wind field on the surface velocity would be and what would these results show.*

The particles were advected by velocities simulated in the uppermost sigma-layer whose thickness did not exceed 1.8 m – we pointed it out in the revised manuscript (Lines 130–133, 242–245). We agree with the reviewer that the velocity field used has a component directly caused by wind stress (i.e. the Ekman wind drift), but this component is unlikely to bias the rotary characteristics of submesoscale eddies in view of huge difference in horizontal length scale between atmospheric cyclones / anticyclones (~1000 km) and submesocale eddies in the ocean (~10 km).

Specific Comments:
*- I would suggest rearranging the introduction and exchanging paragraph line 57-68 with paragraph line 69-79. It seems to me more logical for the structure of the introduction: First, you talk about spirals in general (line 29-38), then about mechanisms how they could arise (line 38-50) and about the modelling of submesoscale structures (line 50-56). If you then take paragraph 69-79 and skip the sentence "As it was mentioned above, a better visualization of the cyclonic spirals is supposedly related to some differences between floating particles rotation in submesoscale cyclonic and anticyclonic eddies which will be investigated hereafter." you will give a clearer reason why to use the Baltic Sea as a study area. Afterwards, the paragraph line 57-68 motivates and presents the objectives of the paper. To conclude the introduction, it would be helpful for the reader to give a short outline of the structure of the paper at the end of the introduction. This would make it easier for the reader to find parts in the paper that are of interest and allows the reader to skip parts they are already familiar with.*

We agree with this remark. The paragraph line 57–68 was moved to the end of Introduction (Lines 103–116).

*- Table 1: Is it necessary to show the whole values in the paper? A table with mean, standard deviation and 95% conf. interval for both anticyclonic and cyclonic eddies could be sufficient for the paper and much more concise. The rest of the table could be shown in the appendix or the supplementary material. Furthermore, all values are also visible in Figure 8.*

Table 1 was moved to Appendix.

*- It would be helpful for the reader if ideas that has been put in brackets as in line 280ff, 309, 311 or 331ff would be outlined in full sentences without brackets to improve the reading flow.*

In the revised manuscript we avoided using brackets in sentences like that of in Line 280 and 309. However, in our opinion, a scientific article differs from fiction in that it is more difficult to read and the reader has to be prepared for this.

*- Discussion and conclusion: I am missing a critical reflection of the sample size of 18 eddies and the choice of the sample: Only data for one summer in one year are chosen. What about other years or seasons? The paper does not need more data yet, but open or further research question could be mentioned in the end of the section.*

A reflection on the sample size and the choice of the sample was added to Chapter 2.2, Lines 251–253, 273–278.

The differences in rotary characteristics of submesoscale cyclonic and anticyclonic eddies were statistically assessed from a limited model output for early summer 2015 in the southeast Baltic Sea, and we could not exclude seasonal and interannual variability of the studied parameters as well as some dependences on the eddy age and lifespan. These issues could be the subject for future research.

The last paragraph was added to the end of Discussion & Conclusion chapter (Lines 464–468).

*Technical Comments:*
*- Could the definition of the eddy radius in line 160-162 also be indicated in Figure 3? It would be easier to understand the definition and why it is a valid definition for this purpose.*

Fig. 3 was designed to explain the definitions of $\omega$ and $r$, the radius and frequency of the particle rotation, and it was not intended / suited to explain the definition of the eddy radius $R$. Meantime, to our mind, verbal definition of $R$ in lines 160–162 (214–216 in the revised manuscript) seems quite clear, unambiguous and constructive: "If a particle is deployed at a large enough distance from the eddy centre, the pseudo-trajectory will inevitably cease to be looped, and the largest $r$ calculated from a still loop-shaped trajectory is taken for eddy radius $R$".

Proposed action: none.

*- Section 2.1: Model setup: What is the temporal resolution of the velocity field?*

The temporal resolution of the velocity field was 10 minutes – the output from the model has been saved with 10 minute resolution for further numerical calculations of particle trajectories. We referred the reader to Väli et al. (2018) for details.

We indicated the temporal resolution in the end of Chapter 2.2 (Lines 242–245).

*- Figure 4-7: Please indicate not only the date but also the exact time as in Figure 3.*

We indicated the exact time in the captions of Figs. 4–7.

**Review #4**

Dear Reviewer #4,

Thank you very much for your comprehensive review of our manuscript. Please find below our replies to your comments. Note that below your comments are written in italic.

*The authors use an ocean model with a very high resolution that evidently is able to resolve a number of fine (sub)mesoscale features. The simulated pattern of eddies fairly well matches the outcome of satellite remote sensing. Most likely this match partially reflects the high probability of having synoptic eddies in certain more or less fixed locations of the Baltic Sea because of the specific geometry of the sea and its shores. Even though this remark is just an observation and not critics, still I recommend making the claim on lines 220-221 a little bit weaker.*

In view of this remark the claim on lines 220–221 was re-written in a weaker formulation as

"The fact that a vortex pair of almost the same size and orientation was modelled in almost the same place and at the same time as the observed vortex pair can be considered as a validation of the model."

*To my eyes, the use of words "linear features" (lines 57, 63 and in several occasions below) is misleading; mostly because in hydrodynamics the adjective "linear" is usually associated with properties of the underlying equations and their solutions. Thus, for many readers "linear surface features" would automatically connote "sinusoidal wave trains" even if Walter Munk used this expression in a different meaning of substances aligned into elongated patches or stripes (like mentioned on line 239).*

Since at the beginning of the manuscript a quote from Munk (1990) with words "linear feature" was given, it would seem inappropriate to completely refuse this term further in the text.

To avoid misleading, we supplemented words "linear features" by "in a tracer field" or "of a tracer concentration".

*I recommend to mention that a "sister" phenomenon of the quick regrouping of particles to cyclonic spirals (lines 302–304; Väli et al., 2018) occurs in the periphery of intense marine eddies. The associated almost explosive increase in the particle concentration in in was first explored in detail in (Samuelson et al., 2012). The increase in the local concentration occurs in the rim of an anticyclonic eddy differently from that in cyclonic ones. It happens basically because of the interaction of outward motions of particles with the field of particles outside the eddy. A little bit outside of the scope of the manuscript is an attempt to quantify the associated systematic changes to the density of particles, with much lower resolution than the simulations in this manuscript, for a subbasin of the Baltic Sea (the Gulf of Finland) in terms of so-called finite-time compressibility (Kalda et al., 2014).*

In view of this remark, we added to the Introduction a short mentioning of (Samuelson et al., 2012) and (Kalda et al., 2014) results as follows (see Lines 78–82).

"Aggregation of simulated floating particles at the edges of anticyclonic eddies as applied to biomass redistribution was explored in (Samuelson et al., 2012). An attempt to quantify the associated systematic changes to the density of particles in terms of so-called finite-time compressibility was made in (Kalda et al., 2014)."

*The entire study, in essence, signals that the well-known asymmetry of atmospheric cyclonic and anticyclonic eddies (all strong storms are cyclonic) becomes evident also in the field of ocean*

*eddies. I guess that the reader would enjoy some comments on whether the established strong asymmetry of the rotation rates of eddies of different sign is a local property (of densely packed eddies?) or reflects a generic property of marine eddies. This asymmetry may affect more widely the statistical parameters of surface flows (Heinloo and Toompuu, 2012) as in such occasions the average curvature of trajectories of water parcels is predominantly of one sign.*

A conspicuous asymmetry of the relative vertical vorticity distribution with a tail of enhanced positive (cyclonic) vorticity values is a generic property of oceanic submesoscale flows (Thomas et al., 2008; McWilliams, 2016) – we pointed it out in the Introduction (Lines 44–49).

In view of this remark we added to Discussion and Conclusions chapter a "semi-intuitive" explanation for strong asymmetry of the rotation rates of eddies of different sign as follows (lines. 429–437).

"The physical intuition for faster spinning of cyclonic eddies vs anticyclonic eddies can be gained from conservation of potential vorticity in a fluid parcel (e.g., Väli et al. (2017): $(\zeta + f)\rho_z = const$, where $\rho_z$ is the vertical gradient of density. If the parcel undergoes ultimate vertical stretching ($\rho_z/\rho_z(0) \to 0$, where $\rho_z(0)$ is the initial value of $\rho_z$) given that it does not spin initially ($\zeta(0) = 0$), it will acquire unlimited cyclonic rotation: $\Omega = \zeta/f = \rho_z(0)/\rho_z - 1 \to \infty$. On the contrary, if the parcel undergoes ultimate vertical squeezing ($\rho_z/\rho_z(0) \to \infty$), it will acquire anticyclonic rotation limited from above: $\Omega \to -1 + 0$. The above considerations make it clear why in Fig. 8 in all cyclonic eddies $\Omega_0 > 1$, while in all anticyclonic eddies except one the rotation speed is within $-1 < \Omega_0 < 0$."

However, the asymmetry of the rotation rates of eddies towards fast spinning cyclonic eddies, to our mind, does not guarantee that the mean vertical vorticity and/or the average curvature of trajectories of water parcels is predominantly positive (cyclonic). This issue deserves a separate study.

*The use of English is clear and appropriate but may need at places minor corrections (e.g. on line 286 it should probably by "the radial distance" but simply "submesoscale cyclones" would do on line 292).*

Thanks, we made the corrections and checked the English once again.

*Minor comments*
*I recommend to be careful with the use of "rotation" of particles and to clearly distinguish rotation of particles around their own centre and (rotary) motion of particles along curved or circular trajectories. For example, the words "floating particles rotation" (line 66) could easily be misinterpreted. Similarly, "the rotation of a particle /—/ is accompanied /—/ by a shift" is ambiguous.*

We changed "particles rotation" for "rotary motion of particles around the centre of eddy" or at least for "rotary motion of particles".

*Some parts of the manuscript contain too long paragraphs that make it complicated to follow the line of thinking. The first paragraph of Introduction covers 27 lines that is far too much. Also, in several occasions the sentences could be split into parts for clarity.*

We split the first paragraph of Introduction and some long sentences.

*Equation (4): it is not clear how w(0) is calculated; also there is no need for square brackets in the first expression.*

To explain how $\omega(0)$ was calculated, we added the following paragraph (Lines 209–214).

"The modelled velocities were bilinearly interpolated to the current position of the particle within the grid cell. Therefore if the initial position of the particle was taken close enough to the exact centre of the eddy, the radius of the loop $r$ would be sufficiently small, e.g. smaller than the grid cell size $dx, dy = 232$ m. The frequency of particle's rotary motion at $r \approx 0.5dx \approx$ 100 m was taken for $\omega(0)$."

The square brackets in Eq. (4) were dropped.

*Line 106: n.m. obviously stands for nautical mile but it is better to explain the abbreviation.*

We changed "n.m." for "nautical mile".

*Line 169: perhaps it would be more exact to speak about divergence/convergence of the surface velocity field.*

We changed divergence(convergence), positive(negative) etc. for divergence/convergence, positive/negative, etc.

*Line 219: use the Polish ´n in Gda´nsk.*

We changed Gdansk for Gdańsk.

*The claim on line 232/233 is just a repetition of the same claim on lines 224-225.*

We dropped the claim on Lines 232–233.

*Table 1 could be better placed in Appendix*

Table 1 was moved to Appendix.

*Line 261: "The statistics : : :" contains, to my eyes, too much jargon and simply "mean" (line 267) would do the same job as "ensemble mean" (but "ensemble mean curve" on line 282 has clear meaning).*

We dropped unnecessary "ensemble" throughout the Line 261 paragraph and in the Fig. 8 caption.